# POLICY CONSISTENCY IN MULTI-AGENT REINFORCEMENT LEARNING WITH MIXED REWARD

## ABSTRACT

The sparsity of team rewards poses a significant challenge that hinders the effective learning of optimal team policies in cooperative multi-agent reinforcement learning. One common approach to mitigate this issue involves augmenting sparse rewards with individual rewards to guide policy training. However, a significant drawback of such approaches is that modifying the reward function can potentially alter the optimal policy. To tackle this challenge, we propose a novel multi-agent policy optimization approach that ensures consistency between the mixed policy (learned from a combination of individual and team rewards) and the team policy (based solely on team rewards), through a new policy consistency constraint that aligns the returns of both policies in policy optimization model. We further develop an iterated policy optimization procedure to solve the formulated problem, deriving an approximate optimization objective for each iteration of the mixed and team policies. Experimental evaluation conducted in the StarCraft II Multi-Agent Challenge Environment (SMAC), Multi-Agent Particle Environment (MPE), and Google Research Football (GRF) environments demonstrate that our proposed approach effectively addresses the policy inconsistency problem, *i.e.*, it evenly outperforms strong baseline methods.

## 1 INTRODUCTION

Cooperative multi-agent reinforcement learning (MARL) has attracted significant interest due to its potential in solving complex decision-making tasks (Yan & Xu, 2020; Chen et al., 2024). Despite advancements in MARL algorithms, the issue of sparse team rewards remains a major obstacle, limiting the practical application of these algorithms in real-world scenarios such as power grids (Tittaferrante & Yassine, 2021), aerial vehicles (Du et al., 2021), and robotics (Sun et al., 2020).

Previous research addressing the issue of sparse rewards commonly relies on additional individual dense rewards. These approaches can be categorized into three main types: utilizing expert knowledge (Kurach et al., 2020; Lowe et al., 2017; Huang et al., 2022; Zhu & Zhao, 2021), exploring the state space (Strehl & Littman, 2008; Bellemare et al., 2016; Liu et al., 2023; Jeon et al., 2022; Liu et al., 2021; Xu et al., 2024), and action exploration (Li et al., 2021; Xu et al., 2023a). While incorporating individual rewards have shown promise in addressing sparse rewards, recent studies highlight a critical issue: learned policies may deviate from optimal policies due to modifications in the reward function, especially in multi-agent environments (Wang et al., 2022). For instance, in a cooperative battle simulation, agents incentivized by individual rewards may prioritize individual skills (such as shooting or escaping) over the collective goal of winning the battle.

The motivation of the our research is illustrated in Figure 1. While team rewards are expected to guide agents towards the optimal policy (orange dashed line), the sparsity of the reward function often hinders the policy learning process (orange solid line). Shaping mixed rewards (through a combination of team and individual rewards) can facilitate more efficient policy learning but may lead to suboptimal policies due to alterations in the reward function (green line). This highlights the need to maintain consistency between the learned policy and the optimal team policy when incorporating mixed rewards.

While IRAT (Wang et al., 2022) mitigates the policy inconsistency issue through improving policy similarity between learned and team policies, our approach completely eliminates this issue by deriving exact policy objectives from a constrained Lagrangian dual optimization model. By maximizing

mixed rewards subject to consistency constraints between learned and team policies' cumulative rewards, we derive an optimization objective with an extended TD error. Unlike IRAT's standard TD error using only individual rewards, CMT incorporates team rewards with a Lagrangian multiplier $\lambda$. Team rewards provide a more comprehensive metric for policy evaluation, while $\lambda$ enforces policy consistency constraints in the dual optimization problem. These innovations yield policies with higher team rewards and reduced variance. Moreover, unlike IRAT's focus on individual rewards, our approach incorporates mixed rewards during training, better balancing individual skill execution and group collaboration.

More specifically, our approach begins with the presented constrained policy optimization problem, which is transformed into its Lagrangian dual form, allowing us to solve it with the unknown optimal team policy. Furthermore, we establish the equivalence between the solutions of the original problem and its dual counterpart.

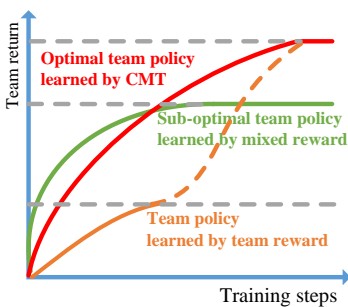

Figure 1: Sparse team rewards often hinders the policy learning process (orange solid line) despite the expectation of guiding agents towards the optimal team policy (orange dashed line). Shaping mixed rewards allows for more efficient policy learning, but it may lead to suboptimal policy due to changes in the reward function (green line). Our approach introduces mixed rewards to efficiently develop policy while ensuring consistency with the optimal team policy (red line). Detailed test results are provided in Section 5.

We propose the Consistency between Mixed and Team policies (CMT) algorithm, which iteratively updates both policies for each agent to solve the Lagrangian dual problem. Using performance difference lemma (Kakade & Langford, 2002) and policy approximation techniques, we simplify the objective function with an extended TD error, while avoiding data inefficiency from simultaneous sampling of both mixed and team policies. Further, we reconstruct the objective with KL terms between policies, maintaining objective equivalence while constraining policy gaps.

Extensive experiments across SMAC, MPE, and GRF environments demonstrate the effectiveness of our proposed approach. Specifically, the proposed approach achieves a 28.5 percentage point higher winning rate and 4.2 percentage point lower standard deviation compared to IRAT across 11 maps of SMAC environments. Furthermore, our approach outperforms other state-of-the-art baselines, including MAPPO, QMIX, MASER, and LAIES, across nearly all tasks. Overall, CMT achieved the best performance in 20 out of 21 tasks across all benchmarks.

## 2 RELATED WORK

**Individual Rewards in MARL:** Introducing individual rewards has become one of the most prevalent and effective strategies to mitigate the sparse reward issue in MARL. Existing research in this area can be broadly categorized into three groups: expert knowledge-based, state space exploration-based, and action space exploration-based approaches.

External expert knowledge plays a crucial role in formulating individual rewards by leveraging prior understanding of environmental dynamics (Kurach et al., 2020; Lowe et al., 2017; Huang et al., 2022; Zhu & Zhao, 2021). For example, a simple design rewards the elimination of enemies and the health of teammates in SMAC (Samvelyan et al., 2019). Further, MAPPER algorithm utilizes expert knowledge to decompose tasks and construct sub-tasks with dense rewards (Liu et al., 2020). However, relying solely on external knowledge can be impractical, as obtaining such knowledge is prohibitively expensive in real-world environments (Zhang et al., 2021a; Ryu et al., 2022).

To address the aforementioned challenge, some works incorporate individual rewards based on acquired transactional information. One straightforward approach is to explore novel states by counting visited states (Strehl & Littman, 2008; Bellemare et al., 2016). However, this approach faces difficulties in complex environments with vast state spaces. Recently, methods such as EDTI/EITI have been developed to promote the exploration of novel states that significantly influence agents' actions (Wang et al., 2019). LAIES partitions the state space into internal and external states, constructing individual rewards to promote exploration of external states (Liu et al., 2023). MASER formulates individual rewards based on the distance between the current state and a target state chosen by the Q-value of visited states and actions (Jeon et al., 2022). Furthermore, DIFFER de-

composes team experience into individual experience for constructing individual rewards (Hu et al., 2023). Additionally, Liu et al. (2021) and Xu et al. (2024; 2023b) exploit prior structural knowledge to encourage agents to explore subsets of the state space.

In contrast to state exploration, another line of research focuses on constructing individual rewards by exploring the action space. One example is the CDS approach (Li et al., 2021), which aims to maximize the mutual information between agent identities and trajectories, thereby encouraging more diverse actions. Xu et al. (2023a) introduce individual rewards based on the concept of joint policy diversity, which quantifies the disparity between the current policy and previous policies.

**Policy consistency in RL:** While previous studies have shown promising results in addressing reward sparsity, the policy inconsistency issue (resulting from the introduction of individual rewards) is often overlooked. To the best of our knowledge, there are two works focusing on this inconsistency issue. The first work proposes a constrained policy optimization method within a single-agent environment (Chen et al., 2022). This method iteratively updates the extrinsic policy and the actual policy using lower bounds of dual objectives as optimization objectives. In contrast, our approach leverages transformed dual optimization objectives to train policies directly, thereby avoiding the bias introduced by lower bounds of the objective function.

The work closest to ours is IRAT (Wang et al., 2022). While IRAT focuses on reshaping the optimization objective to enhance the *policy similarity* between individual and team policies, Figure 4 in Appendix A shows that merely maximizing policy similarity does not necessarily lead the learned policy to converge towards an optimal team policy. Besides, the oversight of mixed rewards prevents IRAT from achieving a balance between individual skill execution and collaborative team objectives. This work therefore derives the optimization objective precisely by solving the Lagrangian dual optimization problem under a policy consistency constraint, which ensures the equivalence of cumulative team rewards obtained by the learned mixed policy and the optimal team policy.

## 3 PRELIMINARIES

In a cooperative multi-agent decision task, the decentralized partially observable Markov decision process (Dec-POMDP) framework defined as $G = <\mathcal{N}, \mathcal{S}, \{\mathcal{A}^i\}_{i\in\mathcal{N}}, P, R, \{\mathcal{Z}^i\}_{i\in\mathcal{N}}, \mathcal{O}, \gamma>$ is commonly used to model the problem. Herein, $\mathcal{N} = \{1, 2, \ldots, n\}$ denotes the set of agents. $\mathcal{S}$ represents the global state space, with $s \in \mathcal{S}$ denoting the environmental state. $\mathcal{A}^i$ is the action space of agent $i$, and $a^i \in \{\mathcal{A}^i\}$ denotes the action taken by agent $i$. $P$ denotes the transition probability, specifying the probability of transitioning from state $s$ to state $s'$ under a joint action $\boldsymbol{a} = (a_t^1, a_t^2, \ldots, a_t^n)$. $R$ represents the reward function, with $r^E \in R$ denoting the shared team rewards received by all agents. $o^i \in \mathcal{Z}^i$ represents the local observation by agent $i$ based on the observation function $\mathcal{O} : \mathcal{S} \times \mathcal{N} \to \mathcal{Z}^i$, where $\mathcal{Z}^i$ denotes the observation space of agent $i$. Given the observation-action history $\tau^i \in T^i = (\{\mathcal{Z}^i\} \times \{\mathcal{A}^i\})$, agent $i$ learns a team policy $\pi_E^i (a^i|\tau^i)$ with the aim to maximize the following **cumulative team reward**:

$$\max J_E \left( \pi_E^i \right) = \mathbb{E}\left[ \sum_{t=0}^{\infty} \gamma^t r_t^E \right], \tag{1}$$

where $\gamma \in [0, 1]$ is the discount factor used to weigh the importance of future rewards. The optimal team policy is defined as $\pi_E^{i\,*} = \arg\max_{\pi_E^i} J_E \left( \pi_E^i \right)$.

Sparse reward presents a common challenge in MARL. To mitigate this issue, individual reward functions are often introduced into policy learning process. In this setting, at each time step $t$, agents select a joint action $\boldsymbol{a}$, and the environment returns a reward $\boldsymbol{r} = (r_t^1, r_t^2, \ldots, r_t^n, r_t^E)$, consisting of individual reward $r_t^i$ and a shared team reward $r_t^E$ for each agent $i$. Consequently, agent $i$ learns a mixed policy $\pi_{E+i}^i$ with the aim to maximize the following **cumulative mixed reward**:

$$\max \hat{J}_{E+i} \left( \pi_{E+i}^i \right) = \mathbb{E}\left[ \sum_{t=0}^{\infty} \gamma^t \left( r_t^E + r_t^i \right) \right], \tag{2}$$

where $\hat{r}_t^i = r_t^E + r_t^i$ is the mixed reward for each agent $i$. Note that reward shaping, including the determination of the importance of individual reward in mixed reward, plays a crucial role in exploration of RL. This is not the primary focus of this paper. For further insights on this topic, readers are referred to Chen et al. (2022) and Yuan et al. (2023).

## 4 METHOD

With a team reward-oriented objective, the optimal team policy for each agent $i$ is denoted as $\pi_E^{i\,*} = \arg\max_{\pi_E^i} J_E\left(\pi_E^i\right)$, as defined in Equation 1. However, when introducing a mixed reward-oriented objective, the optimal mixed policy shifts to $\pi_{E+i}^{i\,*} = \arg\max_{\pi_{E+i}^i} \hat{J}_{E+i}\left(\pi_{E+i}^i\right)$ according to Equation 2. It is evident that, at convergence, the optimal mixed policy must deviate from the optimal team policy due to the change in the reward function.

To resolve this inconsistency issue, we first present a consistency constrained policy optimization model, which defines the research target of our work. Then, we propose a dual policy optimization procedure to solve the optimization model. Finally, we integrate our approach into the centralized training with decentralized execution (CTDE) framework and outline the implementation of our CMT algorithm.

### 4.1 CONSISTENCY CONSTRAINED POLICY OPTIMIZATION MODEL

In the environment incorporating mixed rewards, the policy optimization problem with policy consistency constraint for each agent $i$ is given by:

$$\max_{\pi_{E+i}^i} \hat{J}_{E+i}\left(\pi_{E+i}^i\right) \quad \text{subject to} \quad J_E\left(\pi_{E+i}^i\right) - \max_{\pi_E^i} J_E\left(\pi_E^i\right) = 0. \tag{3}$$

As the performance is often evaluated by the cumulative team rewards $J_E$, the difference between the cumulative team rewards achieved by the learned mixed policy and that learned by the optimal team policy is constrained by Eq. 3. Moreover, the objective function in Eq. 3 remains consistent with the optimization objective introduced in Eq. 2 evaluated by the cumulative mixed rewards $\hat{J}_{E+i}$. As such, the formulated policy optimization problem aims to find a mixed policy that maximizes the mixed rewards while maintaining consistency with the optimal team policy.

To tackle the intractability of directly solving the policy optimization problem with the unknown term $\max_{\pi_E^i} J_E(\pi_E^i)$ in the consistency constraint, we transform the problem into its Lagrangian dual. The Lagrangian dual problem is given by:

$$\min_{\lambda} \left[ \max_{\pi_{E+i}^i} \hat{J}_{E+i}\left(\pi_{E+i}^i\right) + \lambda \left( J_E\left(\pi_{E+i}^i\right) - \max_{\pi_E^i} J_E\left(\pi_E^i\right) \right) \right], \tag{4}$$

where $\lambda$ represents the Lagrangian multiplier associated with the consistency constraint.

To establish the equivalence between the original problem and its Lagrangian dual, we make the following assumption.

**Assumption 1.** *There exists a policy $\pi_{E+i}^i$ such that $J_E\left(\pi_{E+i}^i\right) - \max_{\pi_E^i} J_E\left(\pi_E^i\right) = 0$.*

Assumption 1 requires that there exists a mixed policy $\pi_{E+i}^i$ (developed by the mixed reward in Eq. 2), the performance of which can match that of optimal team policy $\pi_E^{i\,*}$ (defined in Eq. 1) concerning the cumulative team rewards $J_E$. This assumption is commonly observed in RL (Sun & Xu, 2023; Wang et al., 2022). During the initial stage of policy training process, the policy $\pi_{E+i}^i$ yields lower cumulative team reward $J_E$ compared to the optimal team policy $\pi_E^{i\,*}$. However, guided by individual rewards, $\pi_{E+i}^i$ is expected to improve its performance as the training proceeds, and finally approaches or even matches $\pi_E^{i\,*}$. Given this continuous learning process, we can reasonably assume the existence of a policy $\pi_{E+i}^i$ that satisfies $J_E\left(\pi_{E+i}^i\right) - \max_{\pi_E^i} J_E\left(\pi_E^i\right) = 0$.

Under Assumption 1, the Slater's condition (Ding et al., 2020; Zhang et al., 2021b) holds. Consequently, we conclude that the solution to Eq. 3 is equivalent to the solution of its Lagrangian dual problem.

### 4.2 MIN-MAX DUAL POLICY OPTIMIZATION

To find the optimal solution of Eq. 4, we first rewrite it for each agent $i$ as follows:

$$\min_{\lambda} \left[ \min_{\pi_E^i} \max_{\pi_{E+i}^i} \hat{J}_{E+i}^{\lambda}\left(\pi_{E+i}^i\right) - \lambda J_E\left(\pi_E^i\right) \right] \tag{5}$$

where $\hat{J}_{E+i}^{\lambda}\left(\pi_{E+i}^i\right) := \hat{J}_{E+i}\left(\pi_{E+i}^i\right) + \lambda J_E\left(\pi_{E+i}^i\right)$.

In Eq. 5, we observe that there is an opposing optimization objective between the mixed policy $\pi_{E+i}^i$ and the team policy $\pi_E^i$. For the mixed policy $\pi_{E+i}^i$, the objective is to maximize $\hat{J}_{E+i}^{\lambda}\left(\pi_{E+i}^i\right) - \lambda J_E\left(\pi_E^i\right)$, which is minimized in the optimization objective of the team policy $\pi_E^i$. Next, we discuss procedures to optimize the mixed and team policies with the proposed dual objective through an iterated optimization approach.

**Optimizing mixed policy:** The optimizing objective of mixed policy $\pi_{E+i}$ for agent $i$ is

$$\max_{\pi_{E+i}^i} \hat{J}_{E+i}^{\lambda}\left(\pi_{E+i}^i\right) - \lambda J_E\left(\pi_E^i\right). \tag{6}$$

We first expand the objective in Eq. 6 based on the performance difference lemma (Kakade & Langford, 2002) (details are provided in Appendix B.1):

$$\hat{J}_{E+i}^{\lambda}\left(\pi_{E+i}^i\right) - \lambda J_E\left(\pi_E^i\right) = -\mathbb{E}_{\pi_E^i}\left[\sum_{t=0}^{\infty} \lambda r_t^E - V_{E+i}^{\pi_{E+i}^i,\lambda}\left(\tau_t^i\right) + \gamma V_{E+i}^{\pi_{E+i}^i,\lambda}\left(\tau_{t+1}^i\right)\right]$$

$$= -\sum_{\tau^i \in T^i} d_{\rho_0}^{\pi_E^i,\gamma}\left(\tau^i\right) \sum_{a \in A} \pi_E^i\left(a_t^i|\tau_t^i\right) U_{E+i}\left(\tau_t^i, a_t^i\right), \tag{7}$$

where $V_{E+i}^{\pi_{E+i}^i,\lambda} = \mathbb{E}_{\pi_{E+i}^i}\left[\sum_{t=0}^{\infty} \gamma^t \left[r^i + (1+\lambda) r^E\right]\right]$ and $d_{\rho_0}^{\pi_E^i,\gamma}\left(\tau^i\right) = \sum_{t=0}^{\infty} \gamma^t P\left(\tau_t^i = \tau^i|\rho_0, \pi_E^i\right)$.

$U_{E+i}$ is a extended TD error for evaluating mixed policy under the environment with policy consistency requirement. It is defined as:

$$U_{E+i} := \lambda r_t^E - V_{E+i}^{\pi_{E+i}^i,\lambda}\left(\tau_t^i\right) + \gamma V_{E+i}^{\pi_{E+i}^i,\lambda}\left(\tau_{t+1}^i\right). \tag{8}$$

A significant challenge arises from data inefficiency when directly optimizing policies according to Eq. 7, primarily due to the impracticality of sampling from both policies $\pi_{E+i}$ and $\pi_E$ simultaneously. To overcome this challenge, we draw inspiration from the approach proposed in Chen et al. (2022), which leverages trajectories from one policy to approximate another similar policy. Specifically, we approximate the team policy by utilizing trajectories generated by the mixed policy, based on the assumption that mixed and team policies are similar (*similarity assumption*) (Schulman et al., 2017; Kakade & Langford, 2002; Schulman et al., 2015). As a result, the optimization objective can be approximated as follows by changing team policy into mixed policy:

$$-\mathbb{E}_{\pi_E^i}\left[\sum_{t=0}^{\infty} U_{E+i}\left(\tau_t, a_t\right)\right] = -\sum_{\tau^i \in T^i} d_{\rho_0}^{\pi_E^i,\gamma}\left(\tau^i\right) \sum_{a \in A} \pi_{E+i}^i\left(a_t^i|\tau_t^i\right) U_{E+i}\left(\tau_t^i, a_t^i\right)$$

$$= -\sum_{\tau^i \in T^i} d_{\rho_0}^{\pi_E^i,\gamma}\left(\tau^i\right) \sum_{a \in A} \pi_E^i\left(a_t^i|\tau_t^i\right) \frac{\pi_{E+i}^i\left(a_t^i|\tau_t^i\right)}{\pi_E^i\left(a_t^i|\tau_t^i\right)} U_{E+i}\left(\tau_t^i, a_t^i\right). \tag{9}$$

It is worth noting that the *similarity assumption* holds true particularly when the mixed and team policies networks share the same initialized parameters, leading to the minimal disparity between the two types of policies.

To further simplify the computation process while preventing the mixed policy from deviating from the team policy, we introduce a transformation of the objective by incorporating the KL divergence between the mixed and team policies. This transformation maintains the equality between the original objective and the transformed objective. The derivation process for this transformation can be found in Appendix B.1. The final optimization objective for mixed policy is expressed as follows:

$$-\mathbb{E}_{\pi_E^i}\left[\min\left\{\frac{\pi_{E+i}^i\left(a_t^i|\tau_t^i\right)}{\pi_E^i\left(a_t^i|\tau_t^i\right)} U_{E+i}, clip\left(\frac{\pi_{E+i}^i\left(a_t^i|\tau_t^i\right)}{\pi_E^i\left(a_t^i|\tau_t^i\right)}, 1-\epsilon, 1+\epsilon\right) U_{E+i}\right\}\right] - D_{KL}\left(\pi_{E+i}^i||\pi_E^i\right). \tag{10}$$

**Optimizing team policy:** The optimizing objective of team policy $\pi_E^i$ in Eq. 5 can be rewritten as

$$\max_{\pi_E^i} \lambda J_E\left(\pi_E^i\right) - \hat{J}_{E+i}^{\lambda}\left(\pi_{E+i}^i\right). \tag{11}$$

---

**Algorithm 1** CMT algorithm

---

**Input:**
> For each agent $i$, initialize parameters $\theta_{E+i}^i$ for actor network of mixed policy, $\phi_{E+i}^i$ for critic network of mixed policy, $\theta_E^i$ for actor network of team policy, $\phi_E^i$ for critic network of team policy.
> Initialize the Lagrangian multiplier $\lambda$ and learning rate $\alpha$.

1: Initialize empty data buffer $\mathcal{D}_m$ and $\mathcal{D}_t$ for mixed policy and team policy, respectively.
2: **while** $training\ step \leq step_{max}$ **do**
3:    **for** $step = 1, 2, \ldots, step_{max}$ **do**
4:       Initialize empty trajectories lists $\mathcal{D}_m$ and $\mathcal{D}_t$ for mixed policy and team policy, respectively.
5:       Generate mixed action, team action from mixed policy and team policy, respectively.
6:       Interact with environment with mixed action.
7:       Compute the extended TD error $U_{E+i}$ and $U_E$.
8:       Store state, mixed action, reward, termination information and $U_{E+i}$ into $\mathcal{T}_m$.
9:       Store state, team action, reward, termination information and $U_E$ into $\mathcal{T}_t$.
10:     Incorporate $\mathcal{T}_m$ and $\mathcal{T}_t$ into $\mathcal{D}_m$ and $\mathcal{D}_t$, respectively.
11:   **end for**
12:   Sample training data from $\mathcal{D}_m$.
13:   Update actor network of mixed policy according to Eq. 10 and Eq. 29.
14:   Update critic network of mixed policy according to Eq. 31.
15:   Sample training data from $\mathcal{D}_t$.
16:   Update actor network of team policy according to the Eq. 13 and Eq. 30.
17:   Update critic network of team policy according to Eq. 32.
18:   Update Lagrangian multiplier according to Eq. 28.
19: **end while**
**Output:**
> Learned policy $\pi_{E+i}$

---

Similar to the optimization of mixed policy, we transform the optimizing objective as follows (details can be found in Appendix B.2):

$$
\lambda J_E\left(\pi_E^i\right) - \hat{J}_{E+i}^\lambda\left(\pi_{E+i}^i\right) = \mathbb{E}_{\tau_0^i}\left[V_E^{\pi_E^i}\left(\tau_0^i\right)\right] - \lambda\mathbb{E}_{\pi_{E+i}^i}\left[\sum_{t=0}^\infty \gamma^t r_t^{E+i}\right]
$$

$$
= -\mathbb{E}_{\pi_{E+i}^i}\left[\sum_{t=0}^\infty \gamma^t\left((1+\lambda)r_t^E + r_t^i - \lambda V_E^{\pi_E^i}\left(\tau_t^i\right) + \gamma\lambda V_E^{\pi_E^i}\left(\tau_{t+1}^i\right)\right)\right]
$$

$$
:= -\mathbb{E}_{\pi_{E+i}^i}\left[\sum_{t=0}^\infty \gamma^t U_E\left(\tau_t^i, a_t^i\right)\right]. \tag{12}
$$

Based on the same technique of introducing KL divergence term and policy ratio clip during mixed policy optimization, the final optimization objective of team policy is given by:

$$
-\mathbb{E}_{\pi_{E+i}^i}\left[\min\left\{\frac{\pi_E^i\left(a_t^i|\tau_t^i\right)}{\pi_{E+i}^i\left(a_t^i|\tau_t^i\right)}U_E, clip\left(\frac{\pi_E^i\left(a_t^i|\tau_t^i\right)}{\pi_{E+i}^i\left(a_t^i|\tau_t^i\right)}, 1-\epsilon, 1+\epsilon\right)U_E\right\}\right] - D_{KL}\left(\pi_E^i||\pi_{E+i}^i\right). \tag{13}
$$

**Optimizing Lagrangian multiplier:** To update the Lagrangian multiplier $\lambda$, we employ the gradient descent method, considering the optimization objective defined in Eq. 5. The gradient objective for updating $\lambda$ is as follows (cf. Appendix B.3 for the deriving process):

$$
\lambda \leftarrow \lambda - \alpha\mathbb{E}_{\pi_E^i}\left[\sum_{t=0}^\infty \gamma^t \min\left\{\begin{array}{l}\frac{\pi_{E+i}^i\left(a_t^i|\tau_t^i\right)}{\pi_E^i\left(a_t^i|\tau_t^i\right)}A^{\pi_E^i}\left(\tau_t^i, a_t^i\right), \\[2mm] clip\left(\frac{\pi_{E+i}^i\left(a_t^i|\tau_t^i\right)}{\pi_E^i\left(a_t^i|\tau_t^i\right)}, 1-\varepsilon, 1+\varepsilon\right)A^{\pi_E^i}\left(\tau_t^i, a_t^i\right)\end{array}\right\}\right], \tag{14}
$$

where $\alpha$ is the step size, and $A^{\pi_E^i}\left(\tau_t^i, a_t^i\right)$ denotes the TD error for team policy:

$$
A^{\pi_E^i}\left(\tau_t^i, a_t^i\right) = r_t^E + \gamma V_E^{\pi_E^i}\left(\tau_{t+1}^i\right) - V_E^{\pi_E^i}\left(\tau_t^i\right). \tag{15}
$$

Table 1: Median winning rate (%) and standard deviation (%) of five MARL algorithms in more than 10 maps of SMAC environment under rule-based reward setting, using 5 random seeds and at most 10M training steps.

| Map | Difficulty | Rule-based Individual Reward | | | | |
|---|---|---|---|---|---|---|
| | | IRAT | MAPPO | QMIX | Ours | MAPPO (Sparse) |
| 2m_vs_1z | Easy | **100.0(0.0)** | **100.0(0.0)** | **100.0(0.0)** | **100.0(0.0)** | 0.0(0.0) |
| 2s3z | Easy | 98.9(1.1) | 97.2(2.8) | 97.2(2.8) | **100.0(0.0)** | 0.0(0.0) |
| 3m | Easy | **100.0(0.0)** | 97.3(2.3) | 92.6(5.1) | **100.0(0.0)** | 0.0(0.0) |
| 1c3s5z | Easy | 60.9(13.4) | 59.3(13.2) | 98.4(1.6) | **100.0(0.0)** | 0.0(0.0) |
| 3s_vs_5z | Hard | 89.5(4.9) | 87.3(4.5) | 0.0(0.0) | **95.3(1.3)** | 0.0(0.0) |
| 8m_vs_9m | Hard | 56.2(15.6) | 83.2(3.1) | 58.3(24.0) | **93.7(6.2)** | 0.0(0.0) |
| 5m_vs_6m | Hard | 52.6(20.0) | 57.8(8.7) | 62.5(6.3) | **65(6.7)** | 0.0(0.0) |
| 3s5z | Hard | 24.4(22.6) | 64.6(8.8) | 81.0(12.2) | **89.0(9.1)** | 0.0(0.0) |
| MMM2 | Super Hard | 19.1(15.3) | 3.6(2.5) | **71.8(9.9)** | 62.5(22.0) | 0.0(0.0) |
| 6h_vs_8z | Super Hard | 37.5(35.0) | 10.9(6.6) | 49.8(36.1) | **89.0(2.2)** | 0.0(0.0) |
| 3s5z_vs_3s6z | Super Hard | 10.0(0.8) | 34.3(20.6) | 42.4(49.1) | **64.1(35.9)** | 0.0(0.0) |

### 4.3 IMPLEMENTATION

CMT is implemented within the CTDE framework, with MAPPO algorithm as the backbone. The implementation involves two types of policies: mixed policy $\pi_{E+i}^i$ and team policy $\pi_E^i$. Each policy is parameterized by separate networks. During policy execution phase, each agent utilizes the actor network of mixed policy to interact with environment, while the policy information for mixed policy and team policy are stored into data buffer $\mathcal{D}_m$ and $\mathcal{D}_t$, respectively.

When training, CMT algorithm iteratively updates the policies and the Lagrange multiplier $\lambda$. In each iteration, it optimizes the mixed policy $\pi_{E+i}^i$ while keeping the team policy $\pi_E^i$ fixed. It then updates the team policy $\pi_E^i$ based on the optimized mixed policies of all agents. The pseudo-code of CMT algorithm is summarized in Algorithm 1. For more details about the CMT algorithm implementation, please refer to Appendix C.

## 5 EXPERIMENTS

To evaluate the effectiveness of the proposed CMT algorithm, we benchmark it against state-of-the-art baselines across three widely recognized multi-agent benchmarks: SMAC (Samvelyan et al., 2019), MPE (Lowe et al., 2017), and GRF (Kurach et al., 2020). Our approach demonstrates superior performance compared to existing SOTA MARL methods such as IRAT, MAPPO, QMIX, MASER, and LAIES, excelling in 25 out of 27 tasks. Comprehensive experimental details and hyperparameter settings are provided in Appendix D.

### 5.1 EXPERIMENTS ON SMAC

The experiments on SMAC are conducted using two types of individual reward settings:

- **Rule-based Individual Reward**: A sparse reward of 20 is awarded for winning a battle, while a reward of 0 is given otherwise. Additionally, dense individual rewards are allocated to each agent based on the health of team members and enemies. Specifically, an individual reward of 10 is given for each defeated enemy. Meanwhile, a scaled reward is provided according to the agent's remaining health state. Under this reward setting, the CMT algorithm is compared with IRAT (Wang et al., 2022), MAPPO, and the QMIX algorithm. Additionally, MAPPO (Sparse), which trains MAPPO without any individual reward, is included in the experiments to demonstrate the impact of introducing individual rewards. The implementations of IRAT and MAPPO are consistent with the source code in Wang et al. (2022).

- **Heuristic Individual Reward**: The team reward is set at 20 for a battle win. LAIES (Liu et al., 2023) and MASER (Jeon et al., 2022) implement their respective individual rewards, as introduced in Section 2. Since the individual reward in MASER relies on the mixing network of QMIX, which is incompatible with other types of MARL algorithms, our approach employs the same reward setting as LAIES. Under this reward setting, both the proposed and the LAIES algorithms are implemented with IPPO, ensuring all experimental details align with the source code in Liu et al. (2023). The MASER algorithm is implemented with QMIX, using the original source code from Jeon et al. (2022).

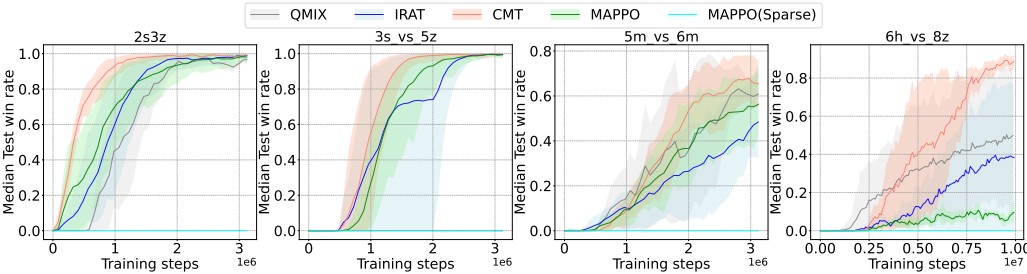

(a) Training curves of five algorithms on four maps of SMAC environments with rule-based individual reward.

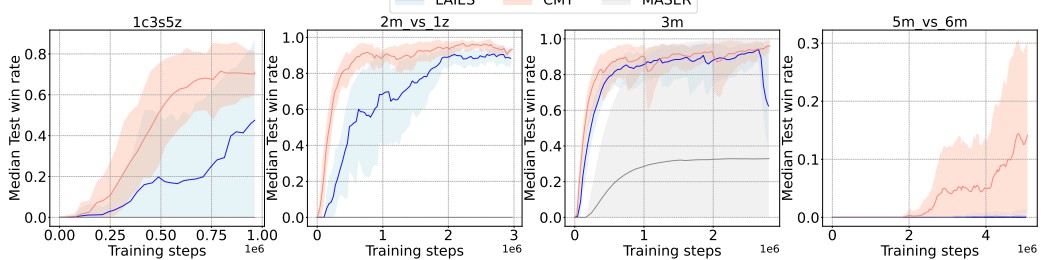

(b) Training curves of three algorithms on four maps of SMAC environments with heuristic individual reward.

Figure 2: Partial training curves on four maps of SMAC environments.

Experimental results for the rule-based individual reward setting are presented in Table 1. As illustrated in Table 1, our approach is the only method that achieves a 100% win rate across all easy maps, and attains the highest win rates on almost all hard and super hard maps. IRAT secures the second-best performance in 6 out of 11 maps. QMIX and MAPPO exhibit varied performances across different maps; In contrast, MAPPO (Sparse), which does not take into account the additional individual rewards, fails to develop an effective policy on any map. These results highlight the significance of introducing effective individual rewards and addressing policy inconsistency when individual rewards are incorporated in sparse reward environments.

We further select four representative maps to display the training curves of five algorithms in Figure 2a. The figures demonstrate that the proposed CMT algorithm exhibits superior sample efficiency compared to the other four algorithms. CMT requires fewer training steps to converge, and its converged win rate is either higher or comparable to other algorithms. Training curves on all 11 maps are provided in Appendix D.3.

Experimental results for the heuristic individual reward setting are presented in Table 2. We evaluated the CMT, MASER, and LAIES algorithms on two easy maps, two hard maps, and one super hard map. As shown in Table 2, our proposed approach achieves the highest win rate across five maps. This result demonstrates that our approach enhances algorithm performance in both rule-based and heuristic individual reward settings. LAIES outperforms or matches MASER in four out of five maps, which is consistent with the results reported in Liu et al. (2023).

Table 2: Median winning rate (%) and standard deviation (%) of three MARL algorithms in 5 maps of SMAC environment under heuristic reward setting, using 5 random seeds and at most 5M training steps.

| Map | Heuristic Individual Reward | | |
|---|---|---|---|
| | MASER | Ours | LAIES |
| 2m_vs_1z | 0.0(0.0) | **100.0(0.0)** | 90.6(5.6) |
| 3m | 63.8(35.3) | **96.8(3.2)** | 82.8(2.2) |
| 1c3s5z | 0.0(0.0) | **72.4(9.9)** | 49.8(43.2) |
| 5m_vs_6m | 15.2(2.8) | **27.1(8.9)** | 0.0(0.0) |
| MMM2 | 0.0(0.0) | **28.2(15.1)** | 15.6(6.8) |

We provide training curves for the three algorithms on four selected maps in Figure 2b. It can be observed that CMT exhibits superior sample efficiency compared to LAIES and MASER. When comparing algorithm performance between heuristic and rule-based individual reward settings, it is evident that the algorithm performance under rule-based individual rewards setting significantly better than that under heuristic individual rewards. This underscores the importance of leveraging environment knowledge and understanding for effective reward shaping.

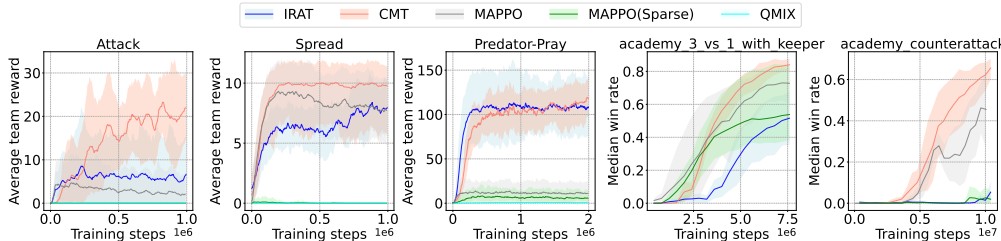

Figure 3: Experimental results on 3 scenarios of MPE and 2 scenarios of GRF

## 5.2 TEST ON MPE

In the MPE environment, we compare our approach with IRAT, MAPPO, QMIX, and MAPPO (Sparse) across three types of environments: Spread, Attack, and Predator-Prey.

In the **Attack environment**, three agents collaborate to reach and attack a single landmark, earning a positive reward 20 upon successful completion. Each agent incurs a penalty based on their distance to the landmark, and a penalty $-1$ when colliding with other agents. As shown in the first column of Figure 3, CMT outperforms all other algorithms, achieving a team reward exceeding 20. IRAT ranks second with a team reward around 7, while MAPPO, QMIX, and MAPPO (Sparse) perform worse, with team rewards below 5. These results indicate that algorithms assisted by individual rewards tend to overlook policy inconsistency, which adversely affects their performance. Although IRAT partially addresses this issue, it fails to achieve consistency between the mixed policy learned from mixed rewards and the team policy derived from team rewards. In contrast, CMT successfully establishes this consistency.

In the **Spread environment**, four agents collaborate to locate two landmarks, receiving a sparse positive reward 10 when multiple agents simultaneously discover a landmark. Additionally, each agent earns an individual reward based on its minimum distance to undiscovered landmarks. Experimental results are presented in the second column of Figure 3. These results demonstrate that CMT once again achieves the highest team rewards, around 10. IRAT and MAPPO follow closely with a team reward of 8. MAPPO (Sparse) and QMIX perform the worst among all algorithms. These findings further show that CMT can identify the sub-optimal policy trap caused by additional individual rewards.

In the **Predator-Prey environment**, five predators work together to capture two prey, receiving a sparse positive reward 20 when multiple agents successfully capture the same prey. Each agent also earns an individual reward 5 when it successfully hits a prey. The results in the Predator-Prey environment differ slightly from those in the previous two environments. As shown in the third column of Figure 3, the CMT and IRAT algorithms perform similarly, with CMT achieving a slightly higher final team reward. This is because in the Predator-Prey environment, individual rewards play a more significant role than in the other two environments. The IRAT algorithm focuses on improving policy performance by fully utilizing individual rewards, whereas the CMT algorithm also considers the consistency between the team policy and the mixed policy.

## 5.3 TEST ON GRF

We also benchmark CMT in the widely-used GRF environment. We employ a common setting where the team reward is defined as $+1$ when the team scores and $-1$ when the team is scored against. The individual reward is based on the default checkpoint, where an agent receives a reward of 0.1 for possessing the football in a region near the goal. We present the results for the two popular academy tasks, $3\_vs\_1$ with the keeper and academy counter-attack (easy) (cf. the experimental settings in Appendix D.4). In the GRF environment, we compare the CMT algorithm with IRAT, MAPPO, and MAPPO (Sparse) algorithms. We did not test QMIX because the GRF environment does not provide global status information.

As shown in the fourth and fifth column of Figure 3, the CMT algorithm achieves the highest win rate, exceeding 60% on both maps. This result demonstrates that CMT can deliver superior performance in environments requiring full collaboration among agents, such as passing and off-ball movement. Even in sparse reward environments, CMT demonstrates excellent performance in developing effective strategies through the utilization of mixed rewards.

Table 3: Average team reward of CMT and five variants, i.e., CMT-TD(w), CMT-SP(w), CMT-KL(w), CMT(3×), CMT(5×) and CMT(RD) on MPE environment using 5 random seeds and at most 2M training steps.

| Scenario | CMT | CMT-TD(w) | CMT-SP(w) | CMT-KL(w) | CMT(3×) | CMT(5×) | CMT(RD) |
|---|---|---|---|---|---|---|---|
| Spread | 9.8(3.9) | 7.2(2.3) | 9.7(2.5) | 7.5(2.5) | 9.5(2.0) | 12.2(1.7) | 0.0(0.0) |
| Attack | 23.7(3.8) | 8.5(4.0) | 13.0(6.0) | 11.0(5.5) | 18.5(6.5) | 17.2(7.5) | 0.0(0.0) |
| Predator-Pray | 112.5(17.5) | 14.0(6.0) | 106.2(26.5) | 22.5(8.0) | 80.5(22.5) | 107.5(20.0) | 7.8(6.0) |

## 5.4 Ablation studies

It is noteworthy that CMT incorporates three crucial design elements: the extended TD error, policy approximation based on the *similarity assumption*, and optimization objective reconstruction with KL divergence. To assess the impact of all components on CMT's performance, we conduct an ablation study in MPE environments. We conduct three ablation studies: i) CMT-TD(w) as CMT with the extended TD error replaced by a standard TD error (defined as Eq. 15) used by MAPPO and IRAT for policy optimization; ii) CMT-SP(w) is CMT with the policy approximation removed by initially using distinct parameters between mixed and team policy networks; iii) CMT-KL(w) represents CMT without the KL terms-based reconstruction module.

Table 3 demonstrates that the inclusion of the extended TD error exerts the most significant impact, followed by the KL divergence-based optimization objective reconstruction model, with the policy approximation technique showing the least influence on algorithm performance. This finding aligns our core innovation outlined in Section 1. The extended TD error, which originates from resolving the policy consistency constraint in the Lagrangian dual problem and the incorporation of mixed rewards, notably contribute to CMT's enhanced performance. Moreover, the minimal disparity between the outcomes of CMT-SP(w) and CMT suggests that eliminating the *similarity assumption* does not dramatically affect algorithm performance.

The influence of scaling individual rewards on the algorithm's performance is also investigated. In Table 3, CMT(3×) denotes the CMT developed with individual rewards amplified three times, while CMT(5×) indicates the CMT developed with individual rewards amplified five times. Comparing CMT(3×) and CMT(5×) with baseline CMT across three scenarios, we observe that the performance variation remains within 30%, indicating CMT's robustness to individual reward scaling.

Finally, we examine CMT's robustness to reward design by testing with random individual rewards sampled from $[-1, 1]$ (denoted as CMT(RD) in Table 3). The results show that with inappropriate individual rewards, CMT(RD) performs similarly to MAPPO(Sparse), both showing limited effectiveness. This indicates that even with poorly designed individual rewards, our approach maintains a performance floor equivalent to methods without additional rewards. Additional ablation studies investigating the impact of the initial Lagrangian multiplier $\lambda$ on CMT's performance can be found in Appendix D.3.

## 6 Conclusion and future work

In this paper, we focus on addressing the challenge of deviation in optimal policies in MARL due to the introduction of individual rewards. To tackle this problem, we propose a novel multi-agent constrained policy optimization procedure, which maximizes the cumulative rewards while ensuring the consistency between the team policy and the mixed policy learned from the sum of team and individual rewards. Leveraging the min-max dual objective presented in the constrained policy optimization model, our approach iteratively updates the mixed and the team policies under the proposed policy consistency constraint. Experimental results validate that CMT successfully overcomes the policy inconsistency issue, gaining superior performance across 27 tasks on MPE, SMAC, and GRF environments, compared to SOTA MARL algorithms.

One limitation of the proposed approach lies that we utilize the *similarity assumption* when simplifying the optimization objectives. Although we can develop policy networks with the same parameters to make this assumption hold during implementation, there may still be divergence between the mixed and team policies during the policy learning process. Relaxing this assumption leads to an interesting direction that is worth further exploration in future work.

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

# APPENDIX

## A VISUAL COMPARISON OF POLICY LEARNING APPROACHES

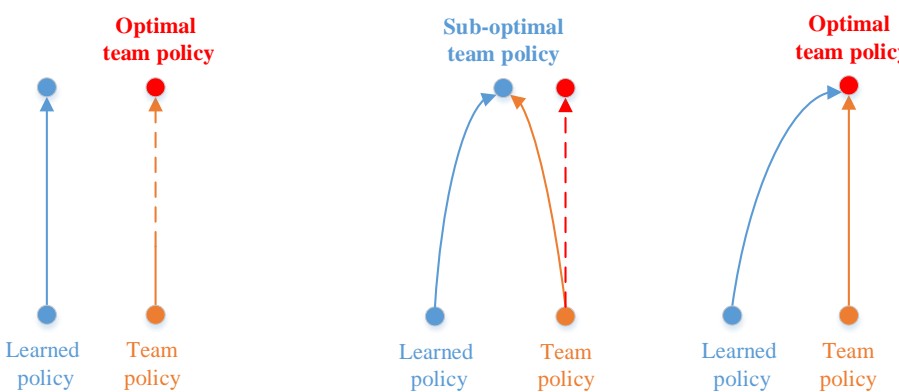

(a) Prior work incorporating individual rewards (e.g., MASER (Jeon et al., 2022), LAIES (Liu et al., 2023))

(b) IRAT (Wang et al., 2022).

(c) Our approach.

Figure 4: Comparison of policy convergence across different approaches. Blue and orange lines represent learned and team policies, respectively. While approaches in (a) and (b) cannot guarantee convergence to the optimal policy, our approach (c) ensures alignment between learned and optimal policies in terms of team rewards.

Figure 4 comprehensively illustrates that our approach fully resolves the policy inconsistency problem compared to existing research. In Figure 4, the learning trajectories of learned and team policies are depicted as blue and orange lines, respectively. In Figure 4a, prior research (e.g., MASER (Jeon et al., 2022) and LAIES (Liu et al., 2023)) incorporates individual rewards into MARL while overlooking the policy inconsistency issue. Consequently, the blue line remains parallel to the orange line, and the optimal team policy fails to develop (shown by dashed lines). In Figure 4b, IRAT mitigates the policy inconsistency problem but does not guarantee convergence of the learned policy to the optimal team policy. As a result, while blue and orange lines gradually approach each other from their starting points and may eventually intersect, the team policy line deviates from its intended direction (red line). In Figure 4c, our approach is the **first** to introduce the consistency policy constraint, enforcing consistency between the mixed policy and optimal team policy's team rewards. Consequently, the direction of team policy remains unchanged, and the blue and orange lines intersect precisely at the optimal team policy point.

## B COMPLETE MATHEMATICAL DERIVATION

### B.1 OPTIMIZATION OBJECTIVE OF MIXED POLICY

The original optimization objective for mixed policy defined in Eq. 6 is:

$$\max_{\pi_{E+i}^i} \hat{J}_{E+i}^\lambda \left( \pi_{E+i}^i \right) - \lambda J_E \left( \pi_E^i \right). \tag{16}$$

To address the data inefficiency issue in sampling policies $\pi_{E+i}$ and $\pi_E$ simultaneously, we make an approximation of optimizing objective in training process of mixed policy network.

First, we expand the objective function as follows:

$$\hat{J}_{E+i}^{\lambda}\left(\pi_{E+i}^{i}\right) - \lambda J_E\left(\pi_E^{i}\right) = \mathbb{E}_{\tau_0}\left[V_{E+i}^{\pi_{E+i}^{i},\lambda}\left(\tau_0^{i}\right)\right] - \lambda\mathbb{E}_{\pi_E^{i}}\left[\sum_{t=0}^{\infty}\gamma^{t}r_t^{E}\right]$$

$$= -\left[\mathbb{E}_{\tau_0^{i}}\left[-V_{E+i}^{\pi_{E+i}^{i},\lambda}\left(\tau_0^{i}\right)\right] + \lambda\mathbb{E}_{\pi_E^{i}}\left[\sum_{t=0}^{\infty}\gamma^{t}r_t^{E}\right]\right] \quad (17)$$

$$= -\mathbb{E}_{\pi_E^{i}}\left[-V_{E+i}^{\pi_{E+i}^{i},\lambda}\left(\tau_0^{i}\right) + \sum_{t=0}^{\infty}\lambda\gamma^{t}r_t^{E}\right].$$

Since the policy $\pi_E^{i}$ has no influence on $V_{E+i}^{\pi_{E+i}^{i},\lambda}$, we can merge the two components in the second equation in Eq. 17, yielding the last equation in Eq. 17.

For simplicity, we let $V_t = V_{E+i}^{\pi_{E+i}^{i},\lambda}\left(\tau_t^{i}\right)$ and $r_t = \lambda r_t^{E}$. We can expand the expression within the expectation in Eq. 16 as follows:

$$-V_0 + \sum_{t=0}^{\infty}\gamma^{t}r_t = (r_0 - V_0 + \gamma V_1) + \gamma(r_1 - V_1 + \gamma V_2) + \gamma^2(r_2 - V_2 + \gamma V_3) + \cdots$$

$$= \sum_{t=0}^{\infty}\gamma^{t}\left(r_t - V_t + \gamma V_t\right)$$

$$= \sum_{t=0}^{\infty}\gamma^{t}\left(\lambda r_t^{E} - V_{E+i}^{\pi_{E+i}^{i},\lambda}\left(\tau_t^{i}\right) + \gamma V_{E+i}^{\pi_{E+i}^{i},\lambda}\left(\tau_{t+1}^{i}\right)\right) \quad (18)$$

$$:= \sum_{t=0}^{\infty}\gamma^{t}U_{E+i}\left(\tau_t^{i}, a_t^{i}\right).$$

Therefore, the optimization objective can be rewritten as:

$$\hat{J}_{E+i}^{\lambda}\left(\pi_{E+i}^{i}\right) - \lambda J_E\left(\pi_E^{i}\right) = -\mathbb{E}_{\pi_E^{i}}\left[\sum_{t=0}^{\infty}\gamma^{t}U_{E+i}\left(\tau_t^{i}, a_t^{i}\right)\right]$$

$$= -\sum_{t=0}^{\infty}\sum_{\tau^{i}\in(\{Z^{i}\}\times\{A^{i}\})}\gamma P\left(\tau_t = \tau|\rho_0, \pi_E^{i}\right)\sum_{a\in A}\pi_E^{i}\left(a_t^{i}|\tau_t^{i}\right)U_{E+i}\left(\tau_t^{i}, a_t^{i}\right)$$

$$= -\sum_{\tau^{i}\in(\{Z^{i}\}\times\{A^{i}\})}d_{\rho_0}^{\pi_E^{i},\gamma}\left(\tau^{i}\right)\sum_{a\in A}\pi_E^{i}\left(a_t^{i}|\tau_t^{i}\right)U_{E+i}\left(\tau_t^{i}, a_t^{i}\right),$$

$$(19)$$

where $d_{\rho_0}^{\pi_E^{i},\gamma}\left(\tau^{i}\right) = \sum_{t=0}^{\infty}\gamma^{t}P\left(\tau_t^{i} = \tau^{i}|\rho_0, \pi_E^{i}\right)$ denotes the discounted observation-action frequency

under policy $\pi_E^{i}$ with the initial observation-action distribution $\rho_0$.

To mitigate the data inefficiency resulting from simultaneously sampling policies $\pi_{E+i}$ and $\pi_E$, as per the *similarity assumption*, we substitute $\pi_E^{i}$ with $\pi_{E+i}^{i}$ and approximate the optimization objectives as follows:

$$\hat{J}_{E+i}^{\lambda}\left(\pi_{E+i}^{i}\right) - \lambda J_E\left(\pi_E^{i}\right) = -\sum_{\tau^{i}\in(\{Z^{i}\}\times\{A^{i}\})}d_{\rho_0}^{\pi_E^{i},\gamma}\left(\tau^{i}\right)\sum_{a\in A}\pi_{E+i}^{i}\left(a_t^{i}|\tau_t^{i}\right)U_{E+i}\left(\tau_t^{i}, a_t^{i}\right). \quad (20)$$

To further simplify the computation process, we introduce the KL divergence between the mixed policy and the team policy while preserving the equality between the original and converted opti-

mization objectives:

$$
\begin{aligned}
\hat{J}_{E+i}^{\lambda}\left(\pi_{E+i}^{i}\right) - \lambda J_{E}\left(\pi_{E}^{i}\right) &= -\left[\sum_{\tau^{i}\in(\{Z^{i}\}\times\{A^{i}\})} d_{\rho_0}^{\pi_E^i,\gamma} \sum_{a\in A} \pi_{E+i}^{i}\left(a_t^i|\tau_t^i\right) U_{E+i}\left(\tau_t^i,a_t^i\right)\right] \\
&\quad + D_{KL}\left(\pi_{E+i}^{i}\parallel\pi_{E}^{i}\right) - D_{KL}\left(\pi_{E+i}^{i}\parallel\pi_{E}^{i}\right) \\
&= -\left[\sum_{\tau^{i}\in(\{Z^{i}\}\times\{A^{i}\})} d_{\rho_0}^{\pi_E^i,\gamma} \sum_{a\in A} \pi_{E}^{i}\left(a_t^i|\tau_t^i\right) \frac{\pi_{E+i}^{i}\left(a_t^i|\tau_t^i\right)}{\pi_{E}^{i}\left(a^i|\tau^i\right)} U_{E+i}\left(\tau_t^i,a_t^i\right)\right] \\
&\quad + D_{KL}\left(\pi_{E+i}^{i}\parallel\pi_{E}^{i}\right) - D_{KL}\left(\pi_{E+i}^{i}\parallel\pi_{E}^{i}\right) \\
&= -\left[\mathbb{E}_{\pi_E}\left[\frac{\pi_{E+i}^{i}\left(a_t^i|\tau_t^i\right)}{\pi_{E}^{i}\left(a_t^i|\tau_t^i\right)} U_{E+i}\left(\tau_t^i,a_t^i\right)\right] - D_{KL}\left(\pi_{E+i}^{i}\|\pi_{E}^{i}\right)\right] \\
&\quad - D_{KL}\left(\pi_{E+i}^{i}\|\pi_{E}^{i}\right).
\end{aligned}
$$
(21)

Leveraging the clip technique presented in Schulman et al. (2017), we finally convert the objective to

$$
\begin{aligned}
\hat{J}_{E+i}^{\lambda}\left(\pi_{E+i}^{i}\right) - \lambda J_{E}\left(\pi_{E}^{i}\right) &= -\mathbb{E}_{\pi_E^i}\left[\min\left\{\begin{array}{l}\frac{\pi_{E+i}^{i}\left(a_t^i|\tau_t^i\right)}{\pi_{E}^{i}\left(a_t^i|\tau_t^i\right)} U_{E+i}\left(\tau_t^i,a_t^i\right), \\ clip\left(\frac{\pi_{E+i}^{i}\left(a_t^i|\tau_t^i\right)}{\pi_{E}^{i}\left(a_t^i|\tau_t^i\right)},1-\epsilon,1+\epsilon\right) U_{E+i}\left(\tau_t^i,a_t^i\right)\end{array}\right\}\right] \\
&\quad - D_{KL}\left(\pi_{E+i}^{i}\|\pi_{E}^{i}\right).
\end{aligned}
$$
(22)

The introduction of the KL-divergence term between the mixed policy and team policy in Eq. 22 not only assists in acquiring the entire optimization objective but also helps to make the *similarity assumption* valid.

### B.2 OPTIMIZATION OBJECTIVE OF TEAM POLICY

The original optimization objective for the team policy, as defined in Eq. 6, is:

$$
\max_{\pi_E^i} \lambda J_E\left(\pi_E^i\right) - \hat{J}_{E+i}^{\lambda}\left(\pi_{E+i}^{i}\right).
$$
(23)

Similar to optimizing the mixed policy, we rewrite the optimization objective for the team policy:

$$
\begin{aligned}
\lambda J_E\left(\pi_E^i\right) - \hat{J}_{E+i}^{\lambda}\left(\pi_{E+i}^{i}\right) &= \mathbb{E}_{\tau_0^i}\left[V_E^{\pi_E^i}\left(\tau_0^i\right)\right] - \lambda\mathbb{E}_{\pi_{E+i}^i}\left[\sum_{t=0}^{\infty}\gamma^t r_t^{E+i}\right] \\
&= -\left[\mathbb{E}_{\tau_0^i}\left[-V_E^{\pi_E^i}\left(\tau_0^i\right)\right] + \lambda\mathbb{E}_{\pi_{E+i}^i}\left[\sum_{t=0}^{\infty}\gamma^t r_t^{E+i}\right]\right] \\
&= -\mathbb{E}_{\pi_{E+i}^i}\left[-V_E^{\pi_E^i}\left(\tau_0^i\right) + \sum_{t=0}^{\infty}\lambda\gamma^t r_t^{E+i}\right] \\
&= -\mathbb{E}_{\pi_{E+i}^i}\left[\sum_{t=0}^{\infty}\gamma^t\left((1+\lambda)r_t^E + r_t^i - \lambda V_E^{\pi_E^i}\left(\tau_t^i\right) + \gamma\lambda V_E^{\pi_E^i}\left(\tau_{t+1}^i\right)\right)\right] \\
&:= -\mathbb{E}_{\pi_{E+i}^i}\left[\sum_{t=0}^{\infty}\gamma^t U_E\left(\tau_t^i,a_t^i\right)\right].
\end{aligned}
$$
(24)

By applying the same technique used in optimizing the mixed policy, we obtain the optimization objective for the team policy:

$$
\lambda J_E\left(\pi_E^i\right) - \hat{J}_{E+i}^\lambda\left(\pi_{E+i}^i\right) = -\mathbb{E}_{\pi_{E+i}^i}\left[\min\left\{\begin{array}{l}\frac{\pi_E^i\left(a_t^i|\tau_t^i\right)}{\pi_{E+i}^i\left(a_t^i|\tau_t^i\right)}U_E\left(\tau_t^i,a_t^i\right),\\ clip\left(\frac{\pi_E^i\left(a_t^i|\tau_t^i\right)}{\pi_{E+i}^i\left(a_t^i|\tau_t^i\right)},1-\epsilon,1+\epsilon\right)U_E\left(\tau_t^i,a_t^i\right)\end{array}\right\}\right]
$$
$$
- D_{KL}\left(\pi_E^i||\pi_{E+i}^i\right).
$$
(25)

### B.3 Optimizing the Lagrangian multiplier $\lambda$

To update Lagrangian multiplier $\lambda$, we start by deriving the objective with respect to $\lambda$ based on the optimization objective defined in Equation 5:

$$
\nabla_\lambda\left(\hat{J}_{E+i}^\lambda\left(\pi_{E+i}^i\right) - \lambda J_E\left(\pi_E^i\right)\right) = J_E\left(\pi_{E+i}^i\right) - J_E\left(\pi_E^i\right).
$$
(26)

To approximate the gradient, we employ a technique presented in the PPO algorithm (Schulman et al., 2017). The gradient can be lower bounded as follows:

$$
J_E\left(\pi_{E+i}^i\right) - J_E\left(\pi_E^i\right) \geq \mathbb{E}_{\pi_E^i}\left[\sum_{t=0}^\infty\gamma^t\min\left\{\begin{array}{l}\frac{\pi_{E+i}^i\left(a_t^i|\tau_t^i\right)}{\pi_E^i\left(a_t^i|\tau_t^i\right)}A^{\pi_E^i}\left(\tau_t^i,a_t^i\right),\\ clip\left(\frac{\pi_{E+i}^i\left(a_t^i|\tau_t^i\right)}{\pi_E^i\left(a_t^i|\tau_t^i\right)},1-\varepsilon,1+\varepsilon\right)A^{\pi_E^i}\left(\tau_t^i,a_t^i\right)\end{array}\right\}\right],
$$
(27)

where the advantage function $A^{\pi_E^i}\left(\tau_t^i,a_t^i\right)$ is defined as $A^{\pi_E^i}\left(\tau_t^i,a_t^i\right) = r_t^E + \gamma V_E^{\pi_E^i}\left(\tau_{t+1}^i\right) - V_E^{\pi_E^i}\left(\tau_t^i\right)$.

Finally, the approximate gradient update step for the Lagrangian multiplier is given by:

$$
\lambda \leftarrow \lambda - \alpha\mathbb{E}_{\pi_E^i}\left[\sum_{t=0}^\infty\gamma^t\min\left\{\begin{array}{l}\frac{\pi_{E+i}^i\left(a_t^i|\tau_t^i\right)}{\pi_E^i\left(a_t^i|\tau_t^i\right)}A^{\pi_E^i}\left(\tau_t^i,a_t^i\right),\\ clip\left(\frac{\pi_{E+i}^i\left(a_t^i|\tau_t^i\right)}{\pi_E^i\left(a_t^i|\tau_t^i\right)},1-\varepsilon,1+\varepsilon\right)A^{\pi_E^i}\left(\tau_t^i,a_t^i\right)\end{array}\right\}\right],
$$
(28)

with $\alpha$ denoting the step size.

## C Implementation Details

### C.1 Auxiliary Objectives

With the MAPPO algorithm as the backbone, CMT adds an auxiliary objective from MAPPO into the entire optimization objectives of mixed policy and policy, respectively.

When updating the mixed policy, an auxiliary objective from MAPPO is added to maximize $\widehat{J}_{E+i}\left(\pi_{E+i}^i\right)$ for agent $i$ as follows:

$$
\max\mathbb{E}\left[\min\left\{\frac{\pi_{E+i}^i\left(a_\tau^i|o_\tau^i\right)}{\pi_{E+i}^{i,old}\left(a_\tau^i|o_\tau^i\right)}A_{E+i}^{i,old}\left(a_\tau^i|o_\tau^i\right),clip\left(\frac{\pi_{E+i}^i\left(a_\tau^i|o_\tau^i\right)}{\pi_{E+i}^{i,old}\left(a_\tau^i|o_\tau^i\right)},1-\epsilon,1+\epsilon\right)A_{E+i}^{i,old}\left(a_\tau^i|o_\tau^i\right)\right\}\right]
$$
(29)

where the advantage function for mixed policy $A^{\pi_{E+i}^i}\left(\tau_t^i,a_t^i\right)$ is defined as $A^{\pi_{E+i}^i}\left(\tau_t^i,a_t^i\right) = \hat{r}_t^i + \gamma V_{E+i}^{\pi_{E+i}^i}\left(\tau_{t+1}^i\right) - V_{E+i}^{\pi_{E+i}^i}\left(\tau_t^i\right)$.

Similarly, when updating the team policy, an auxiliary objective is added to maximize $\widehat{J}_E\left(\pi_E^i\right)$ for agent $i$ as follows:

$$\max \mathbb{E}\left[\min\left\{\frac{\pi_E^i\left(a_\tau^i|o_\tau^i\right)}{\pi_E^{i,old}\left(a_\tau^i|o_\tau^i\right)}A_E^{i,old}\left(a_\tau^i|o_\tau^i\right), clip\left(\frac{\pi_E^i\left(a_\tau^i|o_\tau^i\right)}{\pi_E^{i,old}\left(a_\tau^i|o_\tau^i\right)}, 1-\epsilon, 1+\epsilon\right)A_E^{i,old}\left(a_\tau^i|o_\tau^i\right)\right\}\right]$$

(30)

### C.2 THE UPDATE OF CRITIC NETWORK

The critic network of the mixed policy for agent $i$ is updated in the direction of minimizing the loss function,

$$L\left(\phi_{E+i}^i\right) = \mathbb{E}\left[\max\left[\begin{array}{l}\left(V_{\phi_{E+i}^i}^i\left(\tau_t\right)-R_t^{E+i}\right)^2,\\\left(clip\left(V_{\phi_{E+i}^i}^i\left(\tau_t\right), V_{\phi_{E+i}^i}^i\left(\tau_t\right)-\epsilon, V_{\phi_{E+i}^i}^i\left(\tau_t\right)+\epsilon\right)-R_t^{E+i}\right)^2\end{array}\right]\right],$$

(31)

where $R_t^{E+i}$ denotes the discounted reward-to-go of agent $i$'s mixed reward, which is the cumulative reward obtained by agent $i$ in the mixed policy.

The critic network of team policy for agent $i$ is updated in the direction of minimizing the loss function,

$$L\left(\phi_E^i\right) = \mathbb{E}\left[\max\left[\left(V_{\phi_E^i}^i\left(\tau_t\right)-R_t^E\right)^2, \left(clip\left(V_{\phi_E^i}^i\left(\tau_t\right), V_{\phi_E^i}^i\left(\tau_t\right)-\epsilon, V_{\phi_E^i}^i\left(\tau_t\right)+\epsilon\right)-R_t^E\right)^2\right]\right],$$

(32)

where $R_t^E$ denotes the discounted reward-to-go of agent $i$'s team reward.

## D EXPERIMENT DETAILS

### D.1 CODE BASE

Our approach is implemented with two versions: CMT with MAPPO and CMT with IPPO. When comparing our method with IRAT, MAPPO and QMIX algorithms, since IRAT is implemented with MAPPO algorithm, we implement our approach with MAPPO, and all implementing details are kept consistent with IRAT (Wang et al., 2022). When comparing our method with LAIES and MASER algorithms, since LAIES is implemented with IPPO algorithm, we implement our approach with IPPO, and all implementing details are kept consistent with LAIES (Liu et al., 2023). We sincerely thanks the authors of IRAT and LAIES research for their excellent work producing the codebase.

### D.2 TEST ON MPE ENVIRONMENTS

The experiments were conducted on a computing platform equipped with an AMD Ryzen 9 7950X CPU and a Nvidia 4090 GPU with 96GB of memory. The common parameters for the CMT algorithm and baselines in the MPE environment are summarized in Table 4. The parameters for the IRAT baseline are identical to those reported in Wang et al. (2022). The specific parameters used for the CMT algorithm in the MPE environment are listed in Table 5.

To deeply investigate the CMT algorithm, we conduct more ablation studies to examine the impact of the initial Lagrangian multiplier value selection. Using the Predator-Prey environment of MPE as examples, Figure 5 reveals that the selection of initial values for the Lagrangian multiplier has a significant impact on the algorithm's performance. Notably, positive values of $\lambda$ tend to yield better policy performance compared to negative values of $\lambda$, providing valuable guidance for applying the CMT algorithm in real-world scenarios.

### D.3 TEST ON SMAC ENVIRONMENTS

We employ the SC2.4.10 version as the benchmark to evaluate the performance of all algorithms in the SMAC environment. The parameters of the CMT algorithm on the SMAC environment are provided in Table 6. Full experimental training curves of five algorithms on rule-based individual reward setting on 11 maps are provided in Figure 6, and the training curves of three algorithms on heuristic individual reward setting on 5 maps are provided in Figure 7.

Table 4: The common hyper-parameters of all algorithms on MPE environment.

| Hyper-parameter | Value |
|---|---|
| Number of fully-connected layers | 2 |
| Dim of fully-connected layer | 2 |
| Number of GRU layers | 1 |
| Dim of RNN hidden layer | 64 |
| Optimizer | Adam |
| Value loss | huber loss |
| Huber delta | 10 |
| Batch Size | Number of Envs*Number of Agents*Buffer Length |
| Discount factor $\gamma$ | 0.99 |
| Activation | ReLU |
| Use reward normalization | True |
| Use feature normalization | True |
| Learning rate of Actor Network | 5e-4 |
| Learning rate of Critic Network | 5e-4 |

Table 5: The hyper-parameters of CMT algorithm on MPE environment.

| Hyper-parameter | Value |
|---|---|
| Initial Policy Clipping Ratio in mixed policy | 3.0 |
| Final Policy Clipping Ratio in mixed policy | 0.5 |
| Decaying time range of Policy Clipping Ratio | 2.0 million training steps |
| Policy Clipping Ratio in team policy | 0.2 |
| Learning Rate of Lagrangian multiplier | 0.01 |

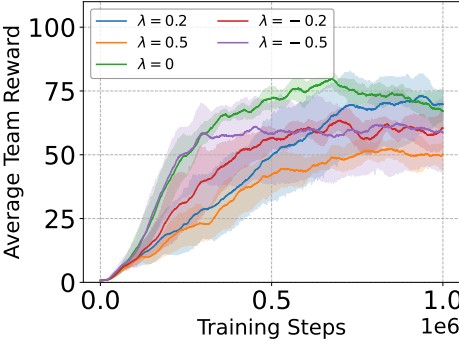

Figure 5: Initial $\lambda$ value ablation

Table 6: The hyper-parameters of CMT algorithm on SMAC environment.

| Hyper-parameter | Value |
|---|---|
| Initial Policy Clipping Ratio in mixed policy | 1.0 |
| Final Policy Clipping Ratio in mixed policy | 0.5 |
| Decaying time range of Policy Clipping Ratio | 0.4 million training steps |
| Policy Clipping Ratio in team policy | 0.05 |
| Learning Rate of Lagrangian multiplier | 0.01 |

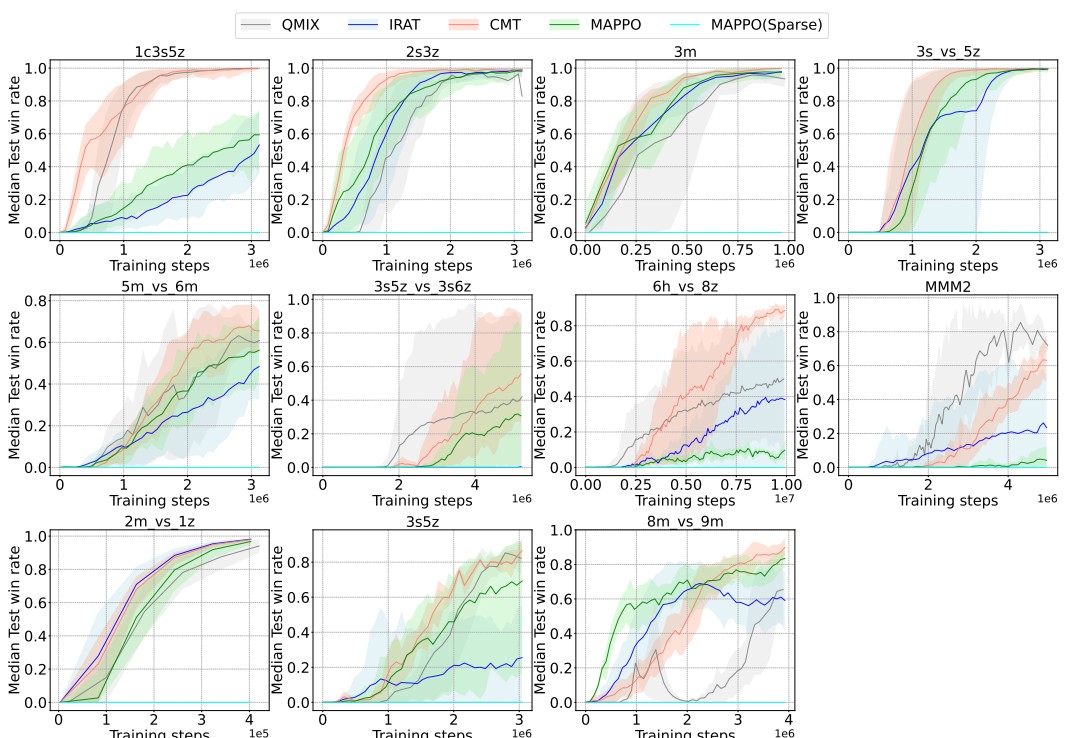

Figure 6: Training curves of five algorithms on rule-based individual reward setting evaluated on 11 maps of SMAC.

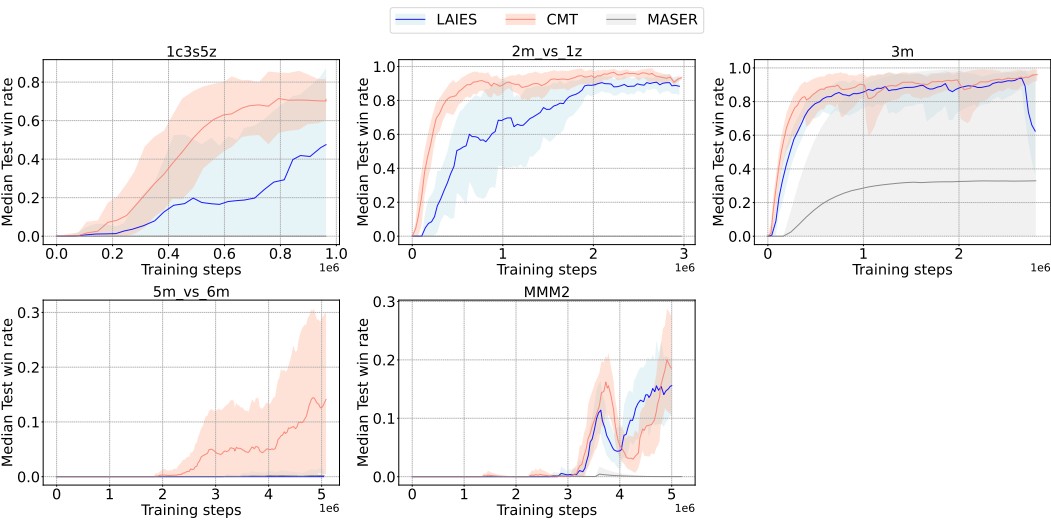

Figure 7: Training curves of three algorithms on heuristic individual reward setting evaluated on 5 maps of SMAC.

## D.4 Test on Football environments

Table 7: The hyper-parameters of CMT algorithm on GRF environment.

| Hyper-parameter | Value |
|---|---|
| Initial Policy Clipping Ratio in mixed policy | 0.5 |
| Final Policy Clipping Ratio in mixed policy | 0.2 |
| Decaying time range of Policy Clipping Ratio | 0.5 million training steps |
| Policy Clipping Ratio in team policy | 0.2 |
| Learning Rate of Lagrangian multiplier | 0.01 |

Under GRF environment, we benchmark the proposed approach on the academy counterattack and academy $3\_vs\_1$ with keeper scenarios. The parameters of the CMT algorithm on the GRF environment are listed in Table 6.

Overall, our experimental results on MPE, SMAC, and GRF environments demonstrate that although the CMT algorithm may not perform the best in the early stages of policy learning, it keeps improving performance as learning progresses. Ultimately, the CMT algorithm achieves the best performance in most benchmarks. These findings demonstrate that the proposed approach can effectively align the learned policy with the optimal policy.

## E  Broader Impact

Our approach, which enables the development of policies that align with the optimal team policy in multi-agent environments with mixed rewards, has far-reaching implications for various real-world applications. For example, in power grid management, unmanned aerial vehicles control, and robotics, where sparse reward functions are prevalent, our approach can be leveraged to efficiently develop policies that ensure consistency with the optimal policy. As such, the proposed approach has the potential to significantly enhance the efficiency and safety performance of MARL algorithms in real-world scenarios.

Furthermore, we are confident that our work has no negative societal implications. Our proposed approach is designed to be benign, with no potential for malicious or unintended uses, and does not raise any concerns related to fairness or privacy.

