# OpenReview forum: "Policy Consistency in Multi-Agent Reinforcement Learning with Mixed Reward"
_ICLR.cc/2025/Conference — Submitted to ICLR 2025_

### Official Review · Reviewer_LvJw · 2024-11-02

**Soundness:** 3
**Presentation:** 3
**Contribution:** 3
**Rating:** 8
**Confidence:** 3

**Summary:**

Augmenting sparse team rewards with individual rewards to guide MARL policy training may potentially alter the optimal policy. This paper proposes a method CMT that ensures consistency between the mixed policy trained with individual and team rewards and the team policy trained solely on team rewards. Specifically, the paper formulates a constrained policy optimization model and transforms it into its Lagrangian dual problem to solve it without knowing the optimal team policy. The authors iteratively optimize mixed policy and team policy with the min-max dual objective derived from the Lagrangian dual model and model them into two types of policy networks. Sufficient experiments are conducted in SMAC, MPE, and GRF to demonstrate CMT’s effectiveness.

**Strengths:**

1. The article is well-structured, with a clear and well-defined motivation.
2. The paper provides a rigorous theoretical derivation to demonstrate the consistency of policy learning after the inclusion of individual rewards.
3. The paper conducted extensive experiments to validate its effectiveness.

**Weaknesses:**

1. The effectiveness of the method relies on the careful design of individual rewards.

**Questions:**

1. According to the training framework proposed by the authors, are there any requirements for individual rewards? In other words, if the rewards are inappropriate, how does the lower bound of this framework compare to the original training method without additional rewards?

2. Based on Question 1, when the scale of individual rewards is too large, even exceeding that of team rewards, will the method in the paper still ensure policy consistency? Can the method eliminate the inconsistency in policy learning objectives caused by the scale disparity between individual rewards and team rewards?

---

> ### Author Response · Authors · 2024-11-20
> **Author Response to Reviewer LvJw**
>
> We sincerely thank Reviewer LvJw for the recognition of our work and for providing constructive comments.
>
> ### **Question 1: If the rewards are inappropriate, how does the lower bound of this framework compare to the original training method without additional rewards?**
>
> **Answer 1**: Thanks for your insightful comments. Following the reviewer’s suggestion, we conducted additional experiments by setting individual rewards randomly from [-1,1] in the MPE environment. These results have been added to Section 5.4 of the revised paper:
>
> |Scenario|CMT|CMT(RD)|MAPPO(Sparse)|
> |:----|:----|:----|:----|
> |Spread|$9.8(3.9)$|$0.0(0.0)$|$0.0(0.0)$|
> |Attack|$23.7(3.8)$|$0.0(0.0)$|$0.0(0.0)$|
> |Predator-Pray|$112.5(17.5)$|$7.8(6.0)$|$6.5(5.5)$|
>
> Here, CMT(RD) represents CMT with random individual rewards. The results demonstrate that with inappropriate individual rewards, CMT(RD) performs similarly to MAPPO(Sparse), both showing limited effectiveness. This suggests that even in worst-case scenarios with poorly designed individual rewards, our approach maintains a performance floor equivalent to methods that operate without additional rewards.
>
> ### **Question 2: When the scale of individual rewards is too large, even exceeding that of team rewards, will the method in the paper still ensure policy consistency?**
>
> **Answer 2**: Thank the reviewer for this valuable question. In Table 3 of the revised manunscript, we added a new ablation study where the individual rewards are amplified three times and five times. Under such settings, the absolute value of individual rewards received in one episode has exceeded the team reward greatly.
>
> |Scenario|CMT|CMT(3$\times$)|CMT(5$\times$)|
> | :----|:----|:----|:----|
> |Spread|$9.8(3.9)$|$9.5(2.0)$|$12.2(1.7)$|
> |Attack|$23.7(3.8)$|$18.5(6.5)$|$17.2(7.5)$|
> |Predator-Pray|$112.5(17.5)$|$80.5(22.5)$|$107.5(20.0)$|
>
> CMT(3×) denotes the CMT developed with individual rewards amplified three times, while CMT(5×) indicates the CMT developed with individual rewards amplified five times. It can be observed that the maximum performance variation remains below 30% across all scenarios. These results demonstrate CMT's robustness to reward magnitude variations while maintaining consistent performance.

---

> > ### Comment · Reviewer_LvJw · 2024-11-21
> >
> > Thank you for the additional experiments, my doubts have been resolved.

---

> > > ### Author Response · Authors · 2024-11-22
> > >
> > > Thank you very much for your positive feedback. We are pleased that the additional experiments have successfully addressed your concerns. We appreciate your time and careful consideration of our manuscript.

---

### Official Review · Reviewer_c1Jb · 2024-11-03

**Soundness:** 3
**Presentation:** 3
**Contribution:** 1
**Rating:** 5
**Confidence:** 4

**Summary:**

This paper investigates the use of individual rewards to enhance team policy learning in cooperative multi-agent reinforcement learning (MARL). The authors introduce a multi-agent policy optimization approach that employs a policy consistency constraint to ensure alignment between the mixed policy and the team policy. The proposed method is evaluated on several benchmarks including SMAC, MPE, and GRF.

**Strengths:**

1. The primary research question - leveraging individual rewards to assist in learning team policies - addresses an important aspect of MARL.
2. The incorporation of a consistency constraint between the returns of the optimal team policy and the learned mixed policy is intuitively sound and well-motivated.

**Weaknesses:**

1. The key innovation of this paper can be further clarified. It's unclear whether the primary contribution is the policy consistency constraint or another aspect. The authors mention IRAT [1] as the most closely related work, but a more detailed comparison between the two methods would be beneficial, particularly in explaining the differences between their respective policy consistency constraints.

2. While this work appears to be an improvement on IRAT, with core methods following similar intuitions of ensuring policy consistency, the reasons for CMT's superior performance over IRAT in the experiments are not thoroughly discussed. An analysis of the key design elements contributing to this improvement would strengthen the paper.

3. The experimental evaluation covers three different benchmark environments (SMAC, MPE, GRF), which is commendable. However, there's inconsistency in the evaluation of baselines across different subsets of tasks. For instance, IRAT, MAPPO, and QMIX are evaluated on 11 tasks in SMAC, while LAIES and MASER are evaluated on 5 tasks. A more uniform evaluation across baselines would provide a clearer comparison.

4. Minor comments:
    - The notation $CD_{KL}$ should be explained upon first use, or the more commonly used $D_{KL}$ could be adopted for clarity.

[1] Wang, Li, et al. "Individual reward assisted multi-agent reinforcement learning." International Conference on Machine Learning. PMLR, 2022.

**Questions:**

1. Could the authors further clarify the main contributions of this paper?
2. What are the key differences between Equation 9 in this paper and Equation 7 in [1]?
3. Could the authors elaborate on the rationale for selecting evaluation tasks?
4. When comparing CMT with LAIES and MASER, does CMT use rule-based individual rewards of LAIES/MASER-constructed individual rewards?

---

> ### Author Response · Authors · 2024-11-20
> **Author Response to Reviewer c1Jb -1/2**
>
> We sincerely thank Reviewer c1Jb for the constructive and valuable comments. The concerns are addressed as follows.
>
> ### **Question 1: Clarify the main contributions of this paper, particularly compared with IRAT.**
>
> **Answer 1**: Thank you for the constructive suggestion, which is helpful in improving the quality of this paper. In the Introduction and Related Work sections of the revised manuscript, we emphasize the main contribution of this paper on the elimination of policy inconsistency, which is also one of the key differences between IRAT and our work.
>
> While IRAT (Wang et al., 2022) mitigates the policy inconsistency issue through improving policy similarity between learned and team policies, our approach completely eliminates this issue by deriving exact policy objectives from a constrained Lagrangian dual optimization model. The standard TD error incorporated in IRAT is based only on individual rewards for policy evaluation, while CMT incorporates team rewards with a Lagrangian multiplier $\lambda$ into TD error. The team reward is a more comprehensive metric (than the individual reward) to evaluate policy performance, and the introduction of Lagrangian multiplier $\lambda$ enforces policy consistency constraints in the dual optimization problem. These innovations yield policies with higher team rewards and reduced variance.
>
> Moreover, unlike IRAT’s focus on individual rewards, our approach incorporates mixed rewards during training, better balancing individual skill execution and group collaboration.
>
> ### **Question 2: An analysis of the key design elements contributing to the performance of CMT would strengthen the paper.**
>
> **Answer 2**: Thank the reviewer for this valuable suggestion. In Section 5.4 of the revised manuscript, we have added ablation studies to analyze the three key components of CMT: extended TD errors ($U_{E+i}$ and $U_{E}$), policy approximation based on similarity assumption, and KL divergence-based optimization objectives reconstruction.
>
> |Scenario|CMT|CMT-TD(w)|CMT-SP(w)|CMT-KL(w)|
> | :---- |:---- | :---- |:----|:----|
> |Spread|$9.8(3.9)$|$7.2(2.3)$|$9.7(2.5)$|$7.5(2.5)$|
> |Attack|$23.7(3.8)$|$8.5(4.0)$|$13.0(6.0)$|$11.0(5.5)$|
> |Predator-Pray|$112.5(17.5)$|$14.0(6.0)$|$106.2(26.5)$|$22.5(8.0)$|
>
> Where i) CMT-TD(w) is CMT with the extended TD error replaced with the standard TD error (defined as Eq. 15) used by MAPPO and IRAT; ii) CMT-SP(w) is CMT with the policy approximation removed by initially using distinct parameters between mixed and team policy networks; iii) CMT-KL(w) represents CMT without the KL terms-based reconstruction module.
>
> The results reveal that the inclusion of the extended TD error exerts the most significant impact, followed by the KL divergence-based optimization objective reconstruction model, with the policy approximation technique showing the least influence on algorithm performance. These findings align with our core contribution outlined in the last answer. The extended TD error, which originates from resolving the policy consistency-constrained Lagrangian dual problem, notably contributes to CMT’s enhanced performance.
>
> ### **Question 3: A more uniform evaluation across baselines when comparing CMT with LAIES and MASER would provide a clearer comparison. Could the authors elaborate on the rationale for selecting evaluation tasks?**
>
> **Answer 3**: Thank you for raising this question. In Table 2, we benchmark CMT against LAIES and MASER in 5 maps of SMAC environment. For the remaining 6 tasks (8m_vs_9m, 3s5z, 3s_vs_5z, 3s5z_vs_3s6z, 2s3z and 6h_vs_8z), both LAIES and MASER achieve a zero reward, and the proposed CMT performs poorly as well. We therefore did not include the numerical results on these 6 tasks.
>
> ### **Question 4: The notation $CD_{KL}$ should be explained upon first use, or the more commonly used $D_{KL}$ could be adopted for clarity.**
>
> **Answer 4**: Thank the reviewer for the constructive suggestion. We have adopted a more common notation $D_{KL}$ in Eqs. (10) and (13) of the revised manuscript.

---

> ### Author Response · Authors · 2024-11-20
> **Author Response to Reviewer c1Jb -2/2**
>
> ### **Question 5: What are the key differences between Equation 9 in this paper and Equation 7 in IRAT[1]?**
>
> **Answer 5**: There are two key differences between the two Equations.
>
> + Firstly, IRAT utilizes a standard TD error, while our approach presents an extended TD error. The standard TD error incorporated in IRAT uses only individual rewards for policy evaluation, while CMT incorporates team rewards with a Lagrangian multiplier $\lambda$ into TD error. The team reward is a better metric (than the individual reward) to evaluate policy performance, and the introduction of the Lagrangian multiplier $\lambda$ represents the policy consistency constraints in the proposed Lagrangian dual problem. These innovative changes enable more accurate policy evaluation under consistency constraints, resulting in policies with higher team rewards and lower variance.
>
> + Further, IRAT utilizes the individual reward to develop the individual policy $\pi_i$ and its state-value function $V$ for each agent $i$, while we utilize the mixed reward to develop the mixed policy $\pi_{E+i}$ and its state-value function $V_{E+i}$. Numerical results show that the use of mixed rewards can better balance individual skill execution and group collaboration.
>
> ### **Question 6: When comparing CMT with LAIES and MASER, does CMT use rule-based individual rewards of LAIES/MASER-constructed individual rewards?**
>
> **Answer 6**: When comparing CMT with LAIES and MASER, our CMT utilizes the individual rewards of LAIES presented. It is a heuristic reward, which is different from the rule-based reward we utilized when comparing CMT with IRAT, QMIX, and MAPPO. In the third paragraph of Section 5.1 of revised manuscript, we have revised the introduction of heuristic individual rewards:
>
> “LAIES (Liu et al., 2023) and MASER (Jeon et al., 2022) implement their respective individual rewards, as introduced in Section 2. Since the individual reward in MASER relies on the mixing network of QMIX, which is incompatible with other types of MARL algorithms, our approach employs the same reward setting as LAIES.”
>
> [1] Li Wang, Yupeng Zhang, Yujing Hu, Weixun Wang, Chongjie Zhang, Yang Gao, Jianye Hao, Tangjie Lv, and Changjie Fan. Individual reward assisted multi-agent reinforcement learning. In International Conference on Machine Learning (ICML), pp. 23417–23432. PMLR, 2022.

---

> > ### Comment · Reviewer_c1Jb · 2024-11-24
> >
> > Thank you to the authors for their response.
> >
> > I want to confirm my understanding: the consistency policy constraint is not a novel contribution of this work but rather a technique adopted from the literature [1]? And the main distinction between this work and [1] lies in the difference between Equation 8 and Equation 15—is that correct?
> >
> > Thank you for clarifying.
> >
> > --------
> > [1] Wang, Li, et al. "Individual reward assisted multi-agent reinforcement learning." International Conference on Machine Learning. PMLR, 2022.

---

> > > ### Author Response · Authors · 2024-11-24
> > > **Further Response to Reviewer c1Jb**
> > >
> > > Thanks for your comments and feedback. Your questions raise an important point regarding our approach.
> > >
> > > To our knowledge, we are the **first** to introduce **the consistency policy constraint** (Eq. 3 of our manuscript) in MARL. Leveraging this innovation, we propose a Lagrangian dual optimization based approach to rigorously design algorithms that maximize the mixed reward while enforcing the consistency between the mixed policy and optimal team policy's team rewards.
> > >
> > > [1] does not incorporate such a consistency policy constraint; instead, [1] adopts a heuristic approach that reshapes the optimization objective directly to improve the policy similarity, which cannot guarantee the satisfaction of policy consistency constraint. We acknowledge that the policy inconsistency issue was presented in [1]. Nevertheless, IRAT proposed in [1] only partially mitigates this issue, while our proposed approach has fully solved this issue.
> > >
> > > IRAT aims to enhance policy similarity between individual and team policies but falls short of developing the learned individual policy converging to the optimal team policy—a common objective in MARL. To illustrate this distinction, consider the learning trajectories of individual and team policies as parallel lines when no additional approach is introduced. In IRAT, these lines gradually approach each other from their starting points, and while they may eventually intersect, the team policy line deviates from its intended direction. Therefore, the learned individual policy fails to converge to the optimal team policy. In contrast, our approach maintains the team strategy's learning direction while guiding the learned mixed strategy to intersect with the team policy at the optimal team policy point, thus comprehensively addressing the policy inconsistency issue. This advantage is empirically validated by our experimental results, which demonstrate CMT's superior performance over IRAT across all 16 evaluation tasks.
> > >
> > > Furthermore, we would like to clarify that the methodological differences between IRAT and CMT extend beyond the distinctions between Eq. 8 and Eq. 15. We have introduced two innovative techniques for deriving optimization objectives:
> > >
> > > 1.	In Eq. 9, we have implemented a network parameters initialization technique to satisfy the similarity assumption, enabling us to use mixed policy trajectories to approximate team policy. This effectively addresses the data inefficiency challenges typically encountered when sampling from both policies. This network initialization technique is introduced in lines 256-258 of the revised manuscript.
> > >
> > > 2.	In Eq. 10, we have implemented an optimization objective reconstruction technique with KL divergence. Unlike simply adding a KL term to the optimization objective, our implemented technique preserves the equality between the original and transformed objectives, thus not undermining the equality between the learned policy's team reward and that of the optimal team policy. This optimization objective reconstruction technique is discussed in lines 259-263 of the revised manuscript.
> > >
> > > Our newly added ablation studies (in Section 5.4 of the revised manuscript) clearly demonstrate the significant contributions of both the network parameter initialization and KL divergence-based optimization objective reconstruction techniques to our approach's effectiveness.
> > >
> > > ***
> > > [1] Wang, Li, et al. "Individual reward assisted multi-agent reinforcement learning." International Conference on Machine Learning. PMLR, 2022.

---

> > > > ### Author Response · Authors · 2024-11-26
> > > >
> > > > Dear reviewer c1Jb,
> > > >
> > > > Thank you for your valuable feedback. To better illustrate the distinction between our approach and existing work, particularly regarding IRAT's [1] treatment of the policy inconsistency issue, we have added Figure 4 in Appendix A of our revised manuscript. This visualization aims to clarify our key contributions. We appreciate your time in reviewing these changes and welcome any additional feedback you may have.
> > > >
> > > > Best regards,
> > > >
> > > > The Authors
> > > >
> > > > ***
> > > > [1] Wang, Li, et al. "Individual reward assisted multi-agent reinforcement learning." International Conference on Machine Learning. PMLR, 2022.

---

> > > > > ### Author Response · Authors · 2024-11-30
> > > > > **Urgent Request for Review Feedback**
> > > > >
> > > > > Dear reviewer c1Jb,
> > > > >
> > > > > Following your question regarding our paper's contribution on November 24th, we have provided two detailed responses: one textual explanation and one figure-based illustration. However, we have not yet received your feedback on these clarifications.
> > > > >
> > > > > We believe our responses comprehensively address your concerns and demonstrate the significant contributions of our work. If you find our explanations satisfactory, we would appreciate your consideration of updating the ratings. If any aspects remain unclear, we welcome your additional questions and will address them promptly before the rebuttal ends.
> > > > >
> > > > > Your insights are invaluable to improving our work.
> > > > >
> > > > > Best regards,
> > > > >
> > > > > Authors

---

> > > > > > ### Comment · Reviewer_c1Jb · 2024-12-03
> > > > > >
> > > > > > Thank you for your clarification, and I sincerely appreciate the authors' efforts in improving the manuscript. Specifically:
> > > > > >
> > > > > > - The original manuscript was somewhat ambiguous regarding the contributions of this work. However, the revised version provides a much clearer motivation that aligns well with the contributions and effectively positions this work within the context of the existing literature.
> > > > > > - The addition of ablation studies enhances the readers’ understanding of how the extended TD error and KL regularization contribute to improving performance.
> > > > > >
> > > > > > I believe the revisions made during the rebuttal period are a step in the right direction, and I encourage the authors to incorporate these changes into the updated manuscript. Based on these improvements, I am willing to adjust my score to a 5, reflecting my optimism about the potential of this work.
> > > > > >
> > > > > > That said, I remain uncertain about whether the contributions of this work, on top of IRAT, are substantial enough to offer significant new insights into mixed reward settings in MARL. Specifically, while the authors provide a new perspective on Lagrangian dual optimization to motivate the policy consistency regularization, the resulting term closely resembles what was proposed in IRAT. Consequently, the novelty of this contribution is somewhat diminished. Based on this, I personally believe that this work, in its current form, falls short of meeting the ICLR acceptance threshold.

---

> > > > > > > ### Author Response · Authors · 2024-12-03
> > > > > > >
> > > > > > > We sincerely appreciate the reviewer's acknowledgment of our revisions and the improved rating. As the reviewer has suggested, we have certainly incorporated the improvements we have made into the revised manuscript.
> > > > > > >
> > > > > > > Regarding the reviewer's concern about our research contributions, we would like to demonstrate how our work offers substantial new insights beyond IRAT into mixed reward settings of MARL through the following aspects:
> > > > > > >
> > > > > > > + IRAT focuses on individual rewards settings and does not explore mixed reward settings. To the best of our knowledge, our work is pioneering in addressing the policy inconsistency issue specifically within the mixed reward setting of MARL, representing a significant advancement in the field.
> > > > > > >
> > > > > > > + We respectfully disagree with the assessment that our resulting term closely resembles IRAT's approach. Despite superficial similarities, our method introduces several fundamental innovations:
> > > > > > >
> > > > > > >   - IRAT's optimization objective (Eq. 5 in [1]) relies on policy similarity definitions (Eq. 1 in [1]) and requires case-by-case analysis for the value of policy similarity. Our approach (Eq. 10 in our manuscript) eliminates the need for such case analysis by directly ensuring equality between the learned policy's team reward and the optimal team policy's reward through our novel policy consistency constraint (Eq. 3 in our manuscript).
> > > > > > >
> > > > > > >   - IRAT directly adds KL divergence between learned and team policies in their optimization objective (Eq. 5 in [1]), potentially compromising reward maximization. Our approach strategically uses KL divergence to reconstruct the optimization objective while maintaining mathematical equivalence with the original objective. This ensures that our final optimization objective (Eq. 10 in our manuscript) effectively guides policy learning toward reward maximization without compromise.
> > > > > > >
> > > > > > >   - Our approach employs extended TD errors (Eq. 8 in our manuscript) and optimization with mixed rewards, in contrast to IRAT's standard TD errors (Last equation in Section 2 of [1]) and individual reward optimization. This fundamental difference leads to substantially different optimization objectives.
> > > > > > >
> > > > > > > Our experimental results substantiate these innovations, demonstrating superior performance compared to IRAT.
> > > > > > >
> > > > > > > Based on these significant distinctions, we believe our work makes substantial contributions to the field of mixed reward settings in MARL, extending well beyond IRAT's framework. We would greatly appreciate the reviewer's reconsideration of the rating in light of these clarifications.
> > > > > > >
> > > > > > > We are deeply grateful for your thorough review and valuable feedback throughout this process.
> > > > > > >
> > > > > > > ***
> > > > > > > [1] Wang, Li, et al. "Individual reward assisted multi-agent reinforcement learning." International Conference on Machine Learning. PMLR, 2022.

---

### Official Review · Reviewer_1Pd6 · 2024-11-04

**Soundness:** 2
**Presentation:** 3
**Contribution:** 2
**Rating:** 5
**Confidence:** 3

**Summary:**

This paper addresses the challenge of deviation in optimal policies in MARL due to the introduction of individual rewards. A multi-agent
constrained policy optimization procedure (CMT), which maximizes the cumulative rewards while ensuring
the consistency between the team policy and the mixed policy, is proposed. Better results are achieved against strong baselines.

**Strengths:**

- The paper is well written and well organized.
- Experimental studies are good.

**Weaknesses:**

- The main weakness, in my opinion, lies in the gap between the actual implementation of CMT and its aim. The paper claims that a significant drawback of previous methods is that modifying the reward function can potentially alter the optimal policy. I was expecting this paper can ensure that the optimization procedure of CMT leads to the optimial team policy. It turns out a number of approximations are made in CMT, and I am not convinded that CMT improves over previous methods in terms of fixing the drawback.

- lack of theoretical results that ensure the optimization procedure of CMT leads to the optimial team policy.

**Questions:**

- Assumption 1. If $\pi_{E+i}^i$ and $\pi_{E}^i$ share the same policy space, Assumption 1 holds by default. I don't understand why this assumption is made explicitly.

- " During policy execution phase, each agent utilizes the actor network of both policies to generate actions and sampling trajectories based on its local information from the environment." Could you please explain this in more details? I don't understand how two polices merge into one execution policy.

---

> ### Author Response · Authors · 2024-11-20
> **Author Response to Reviewer 1Pd6 -1/2**
>
> We thank Reviewer 1Pd6 for the valuable comments. We address the concerns as follows.
>
> ### **Comment 1: There is a gap between the actual implementation of CMT and its aim. The approximations made in CMT cause I am not convinced that CMT improves over previous methods in terms of fixing the policy inconsistency issue.**
>
> **Answer 1**: Thank you for highlighting the gap between the implementation of CMT and its intended goal. In the revised manuscript, we have added discissions to highlight that the approximations (resulting from the similarity assumption and Assumption 1) do not significantly impact CMT's effectiveness in addressing policy inconsistency.
>
> + Firstly, we utilize a key implementation technique to ensure that the "similarity assumption" almost always holds in implementation. We initialize the mixed and team policies networks with the same parameters, contributing to the minimal disparity between the two types of policies.
>
> + Secondly, we have added an ablation study to compare the performance between IRAT and CMT without shared parameter initialization in Section 5.4:
>
>  | Scenario |   CMT |   CMT-SP(w) |  IRAT |
>  | :-----| :---- | :----| :---- |
>  |Spread|$9.8(3.9)$|$9.7(2.5)$| $7.6(2.4)$ |
>  |Attack|$23.7(3.8)$|$13.0(6.0)$| $7.0(5.5)$ |
>  |Predator-Pray|$112.5(17.5)$|$106.2(26.5)$| $105.0(23.5)$ |
>
> The experimental results demonstrate that even without shared initialization (CMT-SP(w)), our approach significantly outperforms IRAT, validating CMT's effectiveness in addressing policy inconsistency.
>
> + Thirdly, Assumption 1 requires that there exists a mixed policy $\pi_{E+i}^i$, the performance of which can match that of optimal team policy ${\pi_E^i}^∗$ concerning the cumulative team rewards $J_E$ . As the reviewer has correctly noted, this assumption naturally holds in our setting where $\pi_{E+i}^i$ and $\pi_E^i$ share the same policy space. We believe that this assumption does not limit CMT's practical effectiveness.
>
> ### **Comment 2: Lack of theoretical results that ensure the optimization procedure of CMT leads to the optimal team policy.**
>
> **Answer 2**: We appreciate your observation regarding theoretical results. While we acknowledge this limitation, the challenge stems from two key factors:
>
> + The inherent complexity of multi-agent environments;
>
> + The general instability of MARL algorithms.
>
> It's worth noting that theoretical guarantees are rare in contemporary MARL algorithms with sparse rewards, including well-established methods like IRAT[1], LAIES[2], and MASER[3]. While we recognize the importance of theoretical foundations, the current state of MARL with sparse rewards often prioritizes empirical validation. We are committed to developing theoretical guarantees in future work.
>
> [1] Li Wang, Yupeng Zhang, Yujing Hu, Weixun Wang, Chongjie Zhang, Yang Gao, Jianye Hao, Tangjie Lv, and Changjie Fan. Individual reward assisted multi-agent reinforcement learning. In International Conference on Machine Learning (ICML), pp. 23417–23432. PMLR, 2022.
>
> [2] Boyin Liu, Zhiqiang Pu, Yi Pan, Jianqiang Yi, Yanyan Liang, and Du Zhang. Lazy agents: a
> new perspective on solving sparse reward problem in multi-agent reinforcement learning. In
> International Conference on Machine Learning (ICML), pp. 21937–21950. PMLR, 2023.
>
> [3] Jeewon Jeon, Woojun Kim, Whiyoung Jung, and Youngchul Sung. Maser: Multi-agent reinforcement learning with subgoals generated from experience replay buffer. In International Conference on Machine Learning (ICML), pp. 10041–10052. PMLR, 2022.

---

> ### Author Response · Authors · 2024-11-20
> **Author Response to Reviewer 1Pd6 -2/2**
>
> ### **Comment 3: Assumption 1. If $\pi_{E+i}$ and $\pi_E^i$ share the same policy space, Assumption 1 holds by default. I don't understand why this assumption is made explicitly.**
>
> **Answer 3**: Thank you for this insightful question. Assumption 1 is essential for two reasons:
>
> + It ensures satisfaction of Slater's condition, which is crucial for establishing the equivalence between the original policy optimization problem and its Lagrangian dual formulation.
>
> + While this assumption naturally holds when $\pi_{E+i}^i$ and $\pi_{E}^i$ share the same policy space, it remains necessary for analytical completeness, particularly when direct proofs are challenging. This approach aligns with similar assumptions in recent work [4].
>
> ### **Comment 4: I don't understand how two policies merge into one execution policy.**
>
> **Answer 4**: Thank you for noting this ambiguity. To clarify: During execution, agents exclusively use the mixed policy for environmental interactions. The team policy is only involved in training but not in actual execution.
>
> We have revised the first paragraph of Section 4.3 to make this distinction explicit: “During policy execution phase, each agent utilizes the actor network of mixed policy to interact with environment."
>
> [4] Yang Zhang, Bo Tang, Qingyu Yang, Dou An, Hongyin Tang, Chenyang Xi, Xueying Li, and Feiyu Xiong. Bcorle (λ): An offline reinforcement learning and evaluation framework for coupons allocation in e-commerce market. Advances in Neural Information Processing Systems, 34:20410–20422, 2021.

---

> > ### Author Response · Authors · 2024-11-30
> > **Urgent Request for Review Feedback**
> >
> > Dear Reviewer 1Pd6,
> >
> > We submitted our response on November 20th but have not yet received your feedback. We believe that the review process is most effective when there is an active dialogue between authors and reviewers, which ultimately benefits both the ICLR conference and the broader RL community.
> >
> > We are confident that our response addresses the concerns you raised and demonstrates substantial improvements to the manuscript. If you find our responses satisfactory, we would greatly appreciate your positive feedback and consideration of updating the ratings. Alternatively, if you have remaining concerns, we welcome your additional feedback and will make every effort to address them within the remaining timeframe before the deadline.
> >
> > Your insights are invaluable to improving our work.
> >
> > Best regards,
> >
> > Authors

---

### Official Review · Reviewer_WpPH · 2024-11-04

**Soundness:** 3
**Presentation:** 2
**Contribution:** 3
**Rating:** 5
**Confidence:** 4

**Summary:**

This work addresses the issue of suboptimal policy deviation when simultaneously utilizing individual rewards and team rewards in multi-agent reinforcement learning (MARL) training. It introduces a consistency constraint and designs an iterative framework for solving the Lagrangian dual problem to derive the optimal policy. Extensive experiments were conducted on several MARL tasks.

**Strengths:**

* The new introduced consistency policy constraint is somehow innovative in addressing the problem targeted by this work and comes with solid theoretical guarantees.
* Extensive experiments were conducted to demonstrate the effectiveness of the proposed method CMT.

**Weaknesses:**

* The presentation of the paper can be improved, especially in the section Introduction and Related Work. Some parts are overly verbose and lack clarity. I hope the authors to further refine their expressions for conciseness and clarity.
* Including the algorithm in the main paper would facilitate readers' understanding of the algorithm.

**Questions:**

* Equation (7) in the main text lacks a definition for $U$. While it is defined in the appendix, it would be better to provide its meaning upon its first appearance in the main text.
* line 216-217, "However, with the help of individual rewards, ... even surpasses $\pi_{E}^{i, *}$". There are some issues with the phrasing here. The use of 'performance' seems to refer to $J_{E+i}$, but it could also be interpreted as $J_{E}$, which does not satisfy the condition for 'surpasses $\pi_{E}^{i, *}$.' I recommend the authors revising this for clarity.

---

> ### Author Response · Authors · 2024-11-20
> **Author Response to Reviewer WpPH**
>
> We thank reviewer WpPH for the valuable and constructive comments. We address the concerns as follows.
>
> ### **Comment 1: The presentation on Introduction and Related work can be improved.**
>
> **Answer 1**: Thank you for your valuable suggestion. We have substantially revised the Introduction and Related Work sections to improve clarity and conciseness. Major improvements include:
>
> 1. Introduction Section:
>
> + Add more discussion to distinguish this work from IRAT, by articulating two key advantages of the proposed approach in Paragraph 4:
>
> $\qquad$① Fully address the policy inconsistency problem by solving a policy consistency-constrained Lagrangian dual optimization problem;
>
> $\qquad$② Balance individual skill execution and group collaboration by incorporating mixed rewards instead of only using team rewards.
>
> + Streamline the technical description in Paragraphs 5-6 to highlight our approach's three key components:
>
> $\qquad$①  Extended TD error derivation using performance difference Lemma;
>
> $\qquad$② Policy approximation techniques;
>
> $\qquad$③ Optimization objective reconstruction with KL terms.
>
> 2. Related Work Section:
>
> + Consolidate similar works [1,2,3] into cohesive thematic groups for more efficient presentation.
>
>
> + Eliminate redundant descriptions, particularly regarding IRAT, which have been covered in the Introduction.
>
> The updated version is also available on OpenReview.
>
> ### **Comment 2: Including the algorithm in the main paper would facilitate readers' understanding of the algorithm.**
>
> **Answer 2**: Thank you for this constructive suggestion. We have incorporated the CMT algorithm's pseudo-code into the main paper. We also added more details about the algorithm implementation to improve readability.
>
> ### **Comment 3: Equation (7) in the main text lacks a definition for U. While it is defined in the appendix, it would be better to provide its meaning upon its first appearance in the main text.**
>
> **Answer 3**: Thank the reviewer for identifying this oversight. We have placed the definition of U at its first appearance in Eq. 7, adding its mathematical formulation and interpretation for clarity.
>
> ### **Comment 4: Clarify what the “performance” refers to and explain about “surpass ${\pi_{E}^i}_{*}$” in Assumption 1.**
>
> **Answer 4**: Thank you for noting this ambiguity. We have clarified that "performance" specifically refers to the cumulative team reward $J_E$. We have also replaced the imprecise term "surpass" with "approaches or even matches" for clarity.
>
> To enhance readability, we have revised the interpretation of Assumption 1 as: " Assumption 1 requires that there exists a mixed policy $\pi_{E+i}^i$ (developed by the mixed reward in Eq. 2), the performance of which can match that of optimal team policy ${\pi _E^i}^*$ (defined in Eq. 1)) concerning the cumulative team rewards $J_E$."
>
> [1] Iou-Jen Liu, Unnat Jain, Raymond A Yeh, and Alexander Schwing. Cooperative exploration for multi-agent deep reinforcement learning. In International Conference on Machine Learning (ICML), pp. 6826–6836. PMLR, 2021.
>
> [2] Pei Xu, Junge Zhang, and Kaiqi Huang. Exploration via joint policy diversity for sparse-reward multi-agent tasks. In Proceedings of the Thirty-Second International Joint Conference on Artificial Intelligence (IJCAI), pp. 326–334, 2023a.
>
> [3] Pei Xu, Junge Zhang, Qiyue Yin, Chao Yu, Yaodong Yang, and Kaiqi Huang. Subspace-aware ex- ploration for sparse-reward multi-agent tasks. In Proceedings of the AAAI Conference on Artificial Intelligence, volume 37, pp. 11717–11725, 2023b.

---

> > ### Author Response · Authors · 2024-11-30
> > **Urgent Request for Review Feedback**
> >
> > Dear Reviewer WpPH,
> >
> > We submitted our response on November 20th but have not yet received your feedback. We believe that the review process is most effective when there is an active dialogue between authors and reviewers, which ultimately benefits both the ICLR conference and the broader RL community.
> >
> > We are confident that our response addresses the concerns you raised and demonstrates substantial improvements to the manuscript. If you find our responses satisfactory, we would greatly appreciate your positive feedback and consideration of updating the ratings. Alternatively, if you have remaining concerns, we welcome your additional feedback and will make every effort to address them within the remaining timeframe before the deadline.
> >
> > Your insights are invaluable to improving our work.
> >
> > Best regards,
> >
> > Authors

---

### Author Response · Authors · 2024-11-20
**General response and revised version of paper**

Dear reviewers,

We sincerely thank all reviewers for their thorough evaluation and valuable feedback. We have addressed each review individually and implemented comprehensive revisions to the manuscript, with key changes highlighted in blue. The major revisions include:

+ [c1Jb] (Section 1) Enhanced presentation of CMT's core contributions.

+ [WpPH] (Sections 1 and 2) Improved the expressions for clarity and conciseness.

+ [c1Jb] (Section 2.2) Expanded comparison between IRAT and CMT.

+ [1Pd6, WpPH] (Section 4.1) Refined motivation and expression of Assumption 1.

+ [WpPH] (Section 4.2) Added the definition and interpretation of extended TD error $U_{E+i}$.

+ [c1Jb] (Section 4.2) Standardized KL term notation.

+ [WpPH] (Section 4.3) Incorporated algorithm pseudo-code.

+ [1Pd6] (Section 4.3) Clarified the execution policy in the implementation of CMT.

+ [c1Jb] (Section 5.1) Explained SMAC heuristic reward settings.

+ [c1Jb, LvJw] (Section 5.1) Extended SMAC experimental results on heuristic individual reward setting.

+ [c1Jb, 1Pd6] (Section 5.4) Added component ablation studies.

+ [LvJw] (Section 5.4) Included individual reward scaling impact analysis.

+ [LvJw] (Section 5.4) Included the lower bound of CMT analysis.

Best,

The Authors

---

### Author Response · Authors · 2024-11-24
**Request for Further Review Feedback**

Dear Reviewers,

We deeply appreciate your time and thoughtful consideration in reviewing our manuscript. We have carefully addressed each point raised in your reviews and have submitted comprehensive responses. As we approach the end of the discussion period, we kindly request any further feedback you may have, as your insights are invaluable in strengthening our work.

If you find our responses satisfactory and our revisions have adequately addressed your concerns, we would be grateful if you would consider raising your rating of our work. On the other hand, should you have any remaining questions or concerns, please do not hesitate to contact us. We are more than willing to provide further clarifications and answer any additional questions you might have.

We sincerely thank you again for your constructive feedback and dedication to the review process.

Best regards,

The Authors

---

### Author Response · Authors · 2024-11-27
**Looking Forward to Further Feedback**

Dear AC and Reviewers,

As the deadline for submitting the revised manuscript approaches, we kindly request your review of our responses to ensure they adequately address your concerns. Your insights would be invaluable for enhancing our final submission.

Once again, we greatly appreciate your comments. We hope to engage in further discussions and hear more feedback from you.

Best regards,

Authors

---

### Meta-Review · Area_Chair_9i1c · 2024-12-21

**Metareview:**

This paper introduces an approach to address policy inconsistency in mixed reward MARL settings. The method combines individual and team rewards while ensuring policy consistency through a constrained optimization framework. While the work demonstrates strong empirical results across multiple benchmarks including SMAC, MPE, and GRF, reviewers expressed varying concerns about its novelty compared to IRAT, theoretical guarantees, and clarity of presentation. Finally, the consensus of the reviews suggests the paper makes valuable contributions but may fall slightly below ICLR's acceptance threshold.

**Additional Comments On Reviewer Discussion:**

The authors provided responses including new ablation studies demonstrating the method's robustness to reward scaling and random rewards, clearer articulation of theoretical foundations, and explicit comparisons with IRAT's approach. While one reviewer was fully satisfied with the responses and maintained a strong positive rating, the other reviewers maintained some reservations despite acknowledging improvements.

---

### Decision · Program_Chairs · 2025-01-22

Reject